# Scalable Cooperative Multi-Agent Reinforcement Learning with Adaptive Communication Range

## Abstract

It has long been recognized that cooperative multi-agent reinforcement learning (MARL) faces scalability challenges, as the state-action space grows exponentially with the number of agents. Existing approaches typically improve scalability by filtering out communication with low-relevance agents, but these approaches often rely on global state and fixed graph distributions, limiting adaptability in large-scale, communication-constrained environments. In this paper, we propose a scalable MARL method, called Adaptive Communication Range PPO (ACR-PPO), that models communication-based decision-making under communication budget constraints as a sequential process: a communication policy first selects each agent's communication range within a given budget, followed by a behavior policy that chooses actions based on the included neighbors. More importantly, we provide a theoretical guarantee of monotonic performance improvement under communication budget constraints. Experimental results across diverse scenarios show that our approach maintains policy performance while significantly reducing communication cost through adaptive range control.

## 1 Introduction

Cooperative multi-agent reinforcement learning (MARL) has achieved notable progress in fields such as game AI (Vinyals et al., 2019), UAV controlling (Cui et al., 2019) and urban transportation (Ghanadbashi & Golpayegani, 2022). However, scaling MARL to large-scale networked systems is challenging due to the exponential expansion of the joint state-action space, resulting in high communication, computational costs, and low sample efficiency (Oroojlooy & Hajinezhad, 2023). Overcoming this scalability challenge is crucial for the practical application of cooperative MARL in real-world environments.

To mitigate the scalability challenges, existing methods can be broadly divided into two categories. The first category of approaches relies on fully decentralized training paradigms, such as IPPO (De Witt et al., 2020) and DPO (Su & Lu, 2024), which extend single-agent techniques to multi-agent settings through parameter sharing. However, due to the absence of inter-agent communication, these approaches often rely on strong assumptions to address the non-stationarity during policy updates. A second category of approaches relaxes communication constraints by allowing agents to interact with local neighbors (Zhang et al., 2021), like DGN (Jiang et al., 2020), Scalable AC (Qu et al., 2020), DMPO (Ma et al., 2024) and Scal-PPO-L (Zhang et al., 2024b). These approaches have significantly improved performance, but they generally rely on predefined communication topologies and require complex hyperparameter tuning, which may limit their flexibility. Some approaches have been proposed to control communication costs—including role-based (Sheng et al., 2022), gating-based approaches (Mao et al., 2020; Wang et al., 2023). These methods typically require the global state, which may limit their effectiveness in large-scale scenarios. Sampling-based methods (Lin et al., 2021; Agorio et al., 2025) are intuitive and relatively easy to implement with a certain degree of flexibility. However, they exhibit limited adaptability and may fail when the underlying distribution changes.

To address the limitations of these approaches, we propose a new cooperative MARL method that formulates communication-based decision-making under communication cost budget as a con-

strained optimization problem. Inspired by HATRPO (Kuba et al., 2022) and ACE (Li et al., 2023a), we model this as a sequential update process (Zhong et al., 2024; Gu et al., 2023), in which the communication policy first dynamically determines each agent's communication range, and then the behavior policy selects actions based on the neighbors within that range.

Our main contributions are summarized as follows:

- We propose a scalable cooperative MARL method that formulates communication-based decision-making under communication cost budget as a constrained optimization problem, which enables adaptive adjustment of the communication range under communication budgets.

- We present the formal definition of the time-varying truncate value function and derive policy optimization objectives for each agent by alternately updating the communication and behavior policies.

- We provide a theoretical guarantee that agents can achieve monotonic performance improvement while satisfying communication budgets constraints.

- We evaluate our approach on the popular networked multi-agent benchmarks, demonstrating that our method maintains competitive policy performance while effectively reducing communication cost by adaptively minimizing the communication range.

## 2 PRELIMINARIES

### 2.1 NETWORKED MULTI-AGENT MDP

We consider a network of $n$ agents represented as an undirected graph $\mathcal{G} = (\mathcal{N}, \varepsilon)$, where $\mathcal{N} = \{1, ..., n\}$ denotes the set of agents, and $\varepsilon$ represents the set of edges capturing the pairwise relationships among them. The networked multi-agent MDP $(\mathcal{G}, \mathcal{S}, \mathcal{A}, \{\mathcal{M}_{ij}\}_{ij \in \varepsilon}, \{r_i\}_{i \in \mathcal{N}})$ are summarized below:

- **State and action**. Each agent $i \in \mathcal{N}$ has a local state $s_i \in \mathcal{S}_i$ and selects an action $a_i \in \mathcal{A}_i$. The global state is defined as $\mathbf{s} = (s_1, s_2, \ldots, s_n)$, where $\mathbf{s} \in \mathcal{S} := \mathcal{S}_1 \times \mathcal{S}_2 \times \cdots \times \mathcal{S}_n$. Similarly, the global action is given by $\mathbf{a} = (a_1, a_2, \ldots, a_n)$, with $\mathbf{a} \in \mathcal{A} := \mathcal{A}_1 \times \mathcal{A}_2 \times \cdots \times \mathcal{A}_n$.

- **Communication**. $\mathcal{M}$ is the message space. $\mathcal{N}_i^\kappa$ is the $\kappa$-hop neighbors of agent $i$. The set of agents not in $\mathcal{N}_i^\kappa$ is denoted by $\mathcal{N}_{-i}^\kappa := \mathcal{N} \setminus \mathcal{N}_i^\kappa$. Each agent obtains the states of all agents within its $\kappa$-hop neighbors $\mathcal{N}_i^\kappa$ through message $m_{ij} \in \mathcal{M}_{ij}$, denoted by $s_{\mathcal{N}_i^\kappa}$.

- **Reward.** Each agent $i$ has a reward function $r_i(s_i, a_i)$. The global reward is defined as the average of individual rewards: $r(\mathbf{s}, \mathbf{a}) = \frac{1}{n} \sum_{i=1}^{n} r_i(s_i, a_i)$.

- **Transition decomposition.** The next state $s_i(t + 1)$ is determined by the $\kappa$-hop neighbors [1] state $s_{\mathcal{N}_i^\kappa}(t)$ and agent $i$'s own action $a_i(t)$ (Qu et al., 2020):

$$P\left(\mathbf{s}(t + 1) | \mathbf{s}(t), \mathbf{a}(t)\right) = \prod_{i=1}^{n} P\left(s_i(t + 1) | s_{\mathcal{N}_i^\kappa}(t), a_i(t)\right). \tag{1}$$

- **Policy factorization.** The joint policy $\psi(\mathbf{a}|\mathbf{s})$ is factorized into individual policies $\psi_i$, each conditioned on the $\kappa$-hop neighbors state $s_{\mathcal{N}_i^\kappa}$ of agent $i$. Formally, $\psi(\mathbf{a}|\mathbf{s}) = \prod_{i \in \mathcal{N}} \psi_i(a_i|s_{\mathcal{N}_i^\kappa})$.

- **Communication cost**. We follow the setting (Er Meng, 2010) and assume that the communication cost grows exponentially with the communication range. For details on the communication cost, see Appendix B.3.

To support the subsequent analysis, we revisit several foundational concepts in MARL. The first key concept is the joint state-action value function $Q$, which is used to estimate the expected cumulative

---

[1]Typically, this $\kappa$ can be any fixed value. This primarily reflects the agents' dependence on state transitions. However, to facilitate subsequent derivations, this paper sets its propagation rate to 1 by default.

discounted reward from a given state-action pair:

$$Q(\mathbf{s}, \mathbf{a}) := \frac{1}{n} \sum_{i=1}^{n} Q_i^{\theta}(\mathbf{s}, \mathbf{a}). \tag{2}$$

More derivation details can be found in the Appendix B.4.

Another key concept is the policy gradient theorem (Schulman et al., 2015)(Lemma 4), which forms the foundation for optimizing the policy objective $J(\theta)$ in many MARL algorithms (Yu et al., 2022; Kuba et al., 2022). Lemma 4 shows that the gradient of $J(\theta)$ depends on joint state-action value function $Q$, which typically involves global dependencies and may limit its scalability in large-scale scenarios. Please refer to the Appendix B.5 for specific details.

## 2.2 SPATIAL CORRELATION DECAY

Spatial correlation decay describes that the impact of agents on each other decays exponentially with their graph distance, which is a commonly made assumption in scalable cooperative MARL (Zhang et al., 2024b; Qu et al., 2020). Following the setting in (Qu et al., 2020), we decompose the state $\mathbf{s}$ into $(s_{\mathcal{N}_i^{\kappa}}, s_{\mathcal{N}_{-i}^{\kappa}})$, where $s_{\mathcal{N}_i^{\kappa}}$ represents the states of agents within the $\kappa$-hop neighbors of agent $i$, and $s_{\mathcal{N}_{-i}^{\kappa}}$ denotes those outside this neighbors. The $(c, \rho)$-exponential decay property is defined as follows:

**Definition 1** *The $(c, \rho)$-exponential decay implies that satisfying $c > 0$ and $\rho = \lambda \in (0, 1)$, the following inequality holds:*

$$|Q_i(s_{\mathcal{N}_i^{\kappa}}, s_{\mathcal{N}_{-i}^{\kappa}}, a_{\mathcal{N}_i^{\kappa}}, a_{\mathcal{N}_{-i}^{\kappa}}) - Q_i(s_{\mathcal{N}_i^{\kappa}}, s'_{\mathcal{N}_{-i}^{\kappa}}, a_{\mathcal{N}_i^{\kappa}}, a'_{\mathcal{N}_{-i}^{\kappa}})| \le c\rho^{\kappa+1}. \tag{3}$$

Based on Definition 1, we introduce the truncated $Q$-functions for agent $i$, which reflect that the dependence on distant agents is "truncated":

$$\widetilde{Q}_i(s_{\mathcal{N}_i^{\kappa}}, s_{\mathcal{N}_{-i}^{\kappa}}) = \sum_{\Lambda} \omega_i(\Lambda) Q_i(s_{\mathcal{N}_i^{\kappa}}, s_{\mathcal{N}_{-i}^{\kappa}}, a_{\mathcal{N}_i^{\kappa}}, a_{\mathcal{N}_{-i}^{\kappa}}), \tag{4}$$

where $\omega_i$ is any non-negative weights,

$$\Lambda = \{s_{\mathcal{N}_i^{\kappa}}, s_{\mathcal{N}_{-i}^{\kappa}}, a_{\mathcal{N}_i^{\kappa}}, a_{\mathcal{N}_{-i}^{\kappa}}\}, \quad \sum_{\Lambda} \omega_i(\Lambda) = 1, \quad \forall (s_{\mathcal{N}_i^{\kappa}}, a_{\mathcal{N}_i^{\kappa}}) \in \mathcal{S}_{\mathcal{N}_i^{\kappa}} \times \mathcal{A}_{\mathcal{N}_i^{\kappa}}.$$

**Lemma 1** *Under the $(c, \rho)$-exponential decay condition, the truncated $Q$-function, as defined in Equation (4), satisfies the following relation:*

$$|Q(s, a) - \widetilde{Q}(s_{\mathcal{N}_i^{\kappa}}, a_{\mathcal{N}_i^{\kappa}})| \le c\rho^{\kappa+1}, \forall (s, a) \in \mathcal{S} \times \mathcal{A}. \tag{5}$$

$$|V(s) - \widetilde{V}(s_{\mathcal{N}_i^{\kappa}})| \le c\rho^{\kappa+1}, \forall (s) \in \mathcal{S}. \tag{6}$$

$$|A(s, a) - \widetilde{A}(s_{\mathcal{N}_i^{\kappa}}, a_{\mathcal{N}_i^{\kappa}})| \le c\rho^{\kappa+1}, \forall (s, a) \in \mathcal{S} \times \mathcal{A}. \tag{7}$$

Lemma 1 shows that, for sufficiently large $\kappa$, spatial correlation decay ensures the truncated $Q$-function is a close approximation to the global $Q$-function. Similarly, the value function $V$ and the advantage function $A$ can also be defined with corresponding properties. The proof of Lemma 1 is provided in Appendix B.2.

Based on Lemma 1 and MAPPO (Yu et al., 2022) loss function, we can derive the "Scalable" MAPPO loss function:

$$L_{actor}(\theta_k) = J(\theta_k) - J(\theta_k^{old})$$
$$= \frac{1}{B} \sum_{i=1}^{B} \sum_{k=1}^{n} \min \left[ \frac{\psi_{\theta_k}(a_k|o_k)}{\psi_{\theta_k^{old}}(a_k|o_k)}, clip\left( \frac{\psi_{\theta_k}(a_k|o_k)}{\psi_{\theta_k^{old}}(a_k|o_k)}, 1 \pm \epsilon \right) \right] \widetilde{A}(s_{\mathcal{N}_i^{\kappa}}, a_{\mathcal{N}_i^{\kappa}}), \tag{8}$$

$$L_{critic}(\theta_k) = \frac{1}{B} \sum_{i=1}^{B} \sum_{k=1}^{n} (\max[(\widetilde{V}(s_{\mathcal{N}_i^{\kappa}}) - R)^2, (clip(\widetilde{V}(s_{\mathcal{N}_i^{\kappa}}), \widetilde{V}(s_{\mathcal{N}_i^{\kappa}}) \pm \epsilon) - R)^2]). \tag{9}$$

Under the assumption of spatial correlation decay, policy updates can rely on $\kappa$-hop local observations $s_{\mathcal{N}_i^\kappa}$ instead of the global state $\mathbf{s}$. However, in existing methods, the communication range $\kappa$ is typically fixed a priori, as it is either set based on global state information or predefined by a static graph structure, thereby limiting adaptability to dynamic and resource-constrained environments such as UAV networks (Yang et al., 2024).

# 3 METHODS

In this section, we introduce time-varying truncation functions to model dynamically communication ranges (Subsection 3.1). Building on this formulation, we address communication-based decision-making under communication cost budget as a constrained optimization problem (Subsection 3.2), and propose a practical algorithm based on parameterized policies (Subsection 3.3).

## 3.1 TIME-VARYING TRUNCATE VALUE FUNCTION

We introduce a time-varying truncated $Q$-value formulation. By leveraging Definition 1 and Lemma 1, we extend the truncation mechanism to define time-varying truncated state-action value function $Q$, advantage function $A$, and along with their corresponding surrogate objectives $L_{\psi_i}$.

**Definition 2** *Let $\rho_i$ denote the state visitation distribution of agent $i$, and let $s_{\mathcal{N}_{i,t}^{\kappa_{i,t}}}$ denote the local state of agent $i$ at timestep $t$ within $\kappa_{i,t}$-hop neighbors. The agent policy $\psi_i$ is factorized as $\psi_i(a_{i,t} \mid s_{\mathcal{N}_{i,t}^{\kappa_{i,t-1}}}) = \phi_i(\kappa_{i,t} \mid s_{\mathcal{N}_{i,t}^{\kappa_{i,t-1}}}) \cdot \pi_i(a_{i,t} \mid s_{\mathcal{N}_{i,t}^{\kappa_{i,t-1}}}, \kappa_{i,t})$, where $\phi_i$ governs the dynamic selection of the communication range $\kappa_{i,t}$, and $\pi_i$ generates the action $a_{i,t}$ given the local state and current communication range. Let $a_{\mathcal{N}_{-i,t}^{\kappa_{i,t}}}$ denote the actions of neighboring agents (excluding agent $i$) within the $\kappa_{i,t}$-hop neighbors.*

*For agent $i$'s communication policy $\phi_i$, we can define:*

$$V_{\psi_i}^{\phi_i}(s_{\mathcal{N}_{i,t}^{\kappa_{i,t-1}}}) = \mathbb{E}_{s \sim \rho_i, a_{i,t} \sim \pi_i, a_{\mathcal{N}_{-i,t}^{\kappa_{i,t}}} \sim \pi_{\mathcal{N}_{-i,t}^{\kappa_{i,t}}}}[Q_{\psi_i}^{\phi_i, \pi_i}(s_{\mathcal{N}_{i,t}^{\kappa_{i,t}}}, a_{i,t}, a_{\mathcal{N}_{-i,t}^{\kappa_{i,t}}})], \quad (10)$$

$$Q_{\psi_i}^{\phi_i}(s_{\mathcal{N}_{i,t}^{\kappa_{i,t-1}}}, \kappa_{i,t}) = r_\kappa + \mathbb{E}_{s \sim \rho_i}[V_{\psi_i}^{\phi_i}(s_{\mathcal{N}_{i,t}^{\kappa_{i,t}}})], \quad (11)$$

$$A_{\psi_i}^{\phi_i}(s_{\mathcal{N}_{i,t}^{\kappa_{i,t-1}}}, \kappa_{i,t}) = Q_{\psi_i}^{\phi_i}(s_{\mathcal{N}_{i,t}^{\kappa_{i,t-1}}}, \kappa_{i,t}) - V_{\psi_i}^{\phi_i}(s_{\mathcal{N}_{i,t}^{\kappa_{i,t-1}}}), \quad (12)$$

$$L_{\psi_i}(\phi_i) = \mathbb{E}_{s \sim \rho_i, \kappa_{i,t} \sim \phi_i}[A_{\psi_i}^{\phi_i}(s_{\mathcal{N}_{i,t}^{\kappa_{i,t-1}}}, \kappa_{i,t})]. \quad (13)$$

*For agent $i$'s behavior policy $\pi_i$, we can define:*

$$Q_{\psi_i}^{\phi_i, \pi_i}(s_{\mathcal{N}_{i,t}^{\kappa_{i,t}}}, a_{i,t}, a_{\mathcal{N}_{-i,t}^{\kappa_{i,t}}}) = r_i + \mathbb{E}_{s \sim \rho_i}[V_{\psi_i}^{\phi_i}(s_{\mathcal{N}_{i,t+1}^{\kappa_{i,t}}})], \quad (14)$$

$$\widetilde{A}_{\psi_i}^{\phi_i, \pi_i}(s_{\mathcal{N}_{i,t}^{\kappa_{i,t-1}}}, \kappa_{i,t}, a_{i,t}, a_{\mathcal{N}_{-i,t}^{\kappa_{i,t}}}) = Q_{\psi_i}^{\phi_i, \pi_i}(s_{\mathcal{N}_{i,t}^{\kappa_{i,t}}}, a_{i,t}, a_{\mathcal{N}_{-i,t}^{\kappa_{i,t}}}) - V_{\psi_i}^{\phi_i}(s_{\mathcal{N}_{i,t}^{\kappa_{i,t}}}), \quad (15)$$

$$L_{\psi_i}(\phi_i, \pi_i) = \mathbb{E}_{s \sim \rho_i, \kappa_{i,t} \sim \phi_i, a_{i,t} \sim \pi_i}[\widetilde{A}_{\psi_i}^{\phi_i, \pi_i}(s_{\mathcal{N}_{i,t}^{\kappa_{i,t-1}}}, \kappa_{i,t}, a_{i,t}, a_{\mathcal{N}_{-i,t}^{\kappa_{i,t}}})]. \quad (16)$$

**Remark.** To satisfy the Markov decision process, the $r_\kappa$ in Equation 11 is not the true reward. This is because, in Definition 3, we also provide the corresponding cost function.

**Proposition 1** *For any agent $i$, suppose the communication range $\kappa_{i,t}$ follows a distribution $\phi_i(s_{\mathcal{N}_{i,t}^{\kappa_{i,t-1}}})$. Let $\Xi = \{s_{\mathcal{N}_{i,t}^{\kappa_{i,t-1}}}, \kappa_{i,t} \sim \phi_i(s_{\mathcal{N}_{i,t}^{\kappa_{i,t-1}}})\}$. Given parameters($\eta = \frac{\bar{r}}{1-\gamma}, \sigma = \lambda$), we can derive:*

$$\sup_{\Xi} |\widetilde{Q}_{\psi_i}^{\phi_i, \pi_i}(s_{\mathcal{N}_{i,t}^{\kappa_{i,t-1}}}, \kappa_{i,t}, a_{i,t}, a_{\mathcal{N}_{-i,t}^{\kappa_{i,t}}}) - Q_{\psi_i}(\mathbf{s}, \mathbf{a})| \leq \mathbb{E}_{\kappa_{i,t} \sim \phi_i(s_{\mathcal{N}_{i,t}^{\kappa_{i,t-1}}})} \eta \sigma^{\kappa_{i,t}+l}. \quad (17)$$

Proposition 1 establishes that varying the communication range $\kappa$ still preserves the spatial decay property of the truncated function. The proof of Proposition 1 is provided in Appendix C.1. Based on this proposition, we can derive the following two corollaries:

**Corollary 1** *For any agent $i$, given parameters $(\eta' = \frac{2\bar{r}+(1-\gamma)G}{1-\gamma}, \sigma = \gamma)$, if Proposition 1 holds, the following result holds:*

$$\sup_{\kappa_{i,t}\sim\phi_i, a_{i,t}\sim\pi_i} |\widetilde{A}_{\psi_i}^{\phi_i,\pi_i}(s_{\mathcal{N}_{i,t}^{\kappa_{i,t-1}}}, \kappa_{i,t}, a_{i,t}, a_{\mathcal{N}_{-i,t}^{\kappa_{i,t}}}) - A_{\psi_i}^{\pi_i}(\boldsymbol{s},\boldsymbol{a}))| \leq \mathbb{E}_{\kappa_{i,t}\sim\phi_i(\cdot|s_{\mathcal{N}_{i,t}^{\kappa_{i,t-1}}})}[\eta'\sigma^{\kappa_{i,t}+l}]. \tag{18}$$

Corollary 1 establishes an upper bound on the error between the time-varying truncated advantage function $\widetilde{A}_{\psi_i}^{\phi_i,\pi_i}$ and the fully centralized advantage function $A_{\psi_i}^{\pi_i}$. This bound shows that the time-varying truncated advantage function can approximate the fully centralized advantage function in policy updates, thereby significantly reducing communication cost. The proof is provided in Appendix C.2.

**Corollary 2** *For any agent $i$, given parameters $(\eta' = \frac{2\bar{r}+(1-\gamma)G}{1-\gamma}, \sigma = \gamma)$, let communication policy $\bar{\phi}_i$ denote the updated policy after selecting the communication range at timestep $t$, and let behavior policy $\bar{\pi}_i$ denote the updated policy after executing the action at timestep $t$. Let $\phi_i^{\text{full}}$ denote the fully centralized communication policy, under which agent $i$ communicates globally at every timestep, i.e., if $\kappa_{i,t} \sim \phi_i^{\text{full}}$, then $s_{\mathcal{N}_{i,t}^{\kappa_{i,t}}} = \boldsymbol{s}$. Combining Equation (16) and Corollary 1, we can obtain:*

$$|L_{\psi_i}(\bar{\phi}_i, \bar{\pi}_i) - L_{\psi_i}(\phi_i^{full}, \bar{\pi}_i)| \leq \mathbb{E}_{\kappa_{i,t}\sim\bar{\phi}_i(\cdot|s_{\mathcal{N}_{i,t}^{\kappa_{i,t-1}}})}[\eta'\sigma^{\kappa_{i,t}+l}]. \tag{19}$$

Corollary 2 establishes an upper bound on the error between the surrogate objective under the time-varying neighbors communication and global communication. This result shows that agents can achieve similar performance in policy updates only using time-varying local neighbors information, thereby significantly reducing communication cost. The proof is provided in Appendix C.3.

For convenience, let $M_{KL} = \nu_{\phi_i} D_{KL}^{\max}(\phi_i, \bar{\phi}_i) + \nu_{\pi_i} D_{KL}^{\max}(\pi_i, \bar{\pi}_i)$ denote a weighted KL divergence penalty, and $L_{\psi_i}(\bar{\psi}_i) = L_{\psi_i}(\bar{\phi}_i) + L_{\psi_i}(\bar{\phi}_i, \bar{\pi}_i)$. We can get:

**Proposition 2** *Given an agent's policy $\psi_i$, it can be decomposed into two components: the communication policy $\phi_i$ and the behavior policy $\pi_i$. The update rules for these policies are as follows:*

$$\bar{\phi}_i = \arg\max_{\phi}(L_{\psi_i}(\hat{\phi}_i) - \nu_{\phi_i} D_{KL}^{\max}(\phi_i, \hat{\phi}_i)), \tag{20}$$

$$\bar{\pi}_i = \arg\max_{\bar{\pi}_i}[L_{\psi_i}(\bar{\phi}_i, \bar{\pi}_i) - \nu_{\pi_i} D_{KL}^{\max}(\pi_i, \bar{\pi}_i) - \mathbb{E}_{\kappa_{i,t}\sim\bar{\phi}_i(\cdot|s_{\mathcal{N}_{i,t}^{\kappa_{i,t-1}}})}[\eta'\rho^{\kappa_{i,t}+l}]. \tag{21}$$

*Correspondingly,*

$$\nu_{\phi_i} = \frac{4\gamma \max_{s,\kappa_{i,t}} |A_{\psi_i}^{\phi_i}(s_{\mathcal{N}_{i,t}^{\kappa_{i,t-1}}}, \kappa_{i,t})|}{(1-\gamma)^2}, \nu_{\pi_i} = \frac{4\gamma \max_{s,\kappa_{i,t},a_{i,t}} |\widetilde{A}_{\psi_i}^{\phi_i,\pi_i}(s_{\mathcal{N}_{i,t}^{\kappa_{i,t-1}}}, \kappa_{i,t}, a_{i,t}, a_{\mathcal{N}_{-i,t}^{\kappa_{i,t}}})|}{(1-\gamma)^2}.$$

*Under the constraints imposed by $D_{KL}^{\max}(\phi_i, \hat{\phi}_i)$ and $D_{KL}^{\max}(\pi_i, \hat{\pi}_i)$, the updated policies $\bar{\phi}_i$ and $\bar{\pi}_i$ are encouraged to remain close to $\phi_i$ and $\pi_i$, respectively. The updated joint policy $\bar{\psi}_i$ is thus obtained by solving the constrained optimization problem:*

$$J(\bar{\psi}_i) - J(\psi_i) \geq L_{\psi_i}(\bar{\psi}_i) - \mathbb{E}_{\kappa_{i,t}\sim\bar{\phi}_i(\cdot|s_{\mathcal{N}_{i,t}^{\kappa_{i,t-1}}})}[\eta'\rho^{\kappa_{i,t}+l}] - M_{KL}. \tag{22}$$

Proposition 2 shows that the communication policy and the behavior policy can be jointly optimized, enabling agents to make decisions based only on communication within their $\kappa$-hop neighbors. The proof is provided in Appendix C.4.

## 3.2 SCALABLE MARL WITH COMMUNICATION RANGE

Although the performance of policy $\psi_i$ can be guaranteed to monotonically improve under time-varying conditions, it tends to select larger communication ranges for improving performance. This phenomenon forces us to ensure performance improvement under the upper bound on communication cost.

**Definition 3** *Given an agent's policy $\psi_i$, it decomposes into two components: the communication range control policy $\phi_i$ and the beharior policy $\pi_i$. The cost constraint induced by $\phi_i$ is defined as follows:*

$$J_c(\psi_i) = \mathbb{E}_{s \sim \rho_i, \kappa_{i,t} \sim \phi_i(\cdot | s_{\mathcal{N}_{i,t}^{\kappa_{i,t-1}}})}[\gamma^t C(s_{\mathcal{N}_{i,t}^{\kappa_{i,t-1}}}, \kappa_{i,t})] \leq c_i. \tag{23}$$

*The surrogate objective under the communication cost constraint is given by:*

$$L_{c,\psi_i}(\bar{\phi}_i) = \mathbb{E}_{s \sim \rho_i, \kappa_{i,t} \sim \bar{\phi}_i(\cdot | s_{\mathcal{N}_{i,t}^{\kappa_{i,t-1}}})}[A_{c,\psi_i}^{\bar{\phi}_i}(s_{\mathcal{N}_{i,t}^{\kappa_{i,t-1}}}, \kappa_{i,t})]. \tag{24}$$

Similar to Equation (8), we can compute how the communication cost changes as agents update their communication policy:

**Lemma 2** *Let $\psi_i$ and $\bar{\psi}_i$ denote the old and new policies of agent $i$, respectively. Similarly, $\phi_i$ and $\bar{\phi}_i$ denote the old and new communication policies of agent $i$. The following inequality holds:*

$$J_c(\bar{\psi}_i) - J_c(\psi_i) \leq L_{c,\psi_i}(\bar{\phi}_i) - \frac{4\gamma \max_{s,\kappa_{i,t}} |A_{c,\psi_i}^{\bar{\phi}_i}(s_{\mathcal{N}_{i,t}^{\kappa_{i,t-1}}}, \kappa_{i,t})|}{(1-\gamma)^2} D_{KL}^{\max}(\phi_i, \bar{\phi}_i). \tag{25}$$

Lemma 2 shows that, when the divergence between $\phi_i$ and $\bar{\phi}_i$ is sufficiently small, the change in communication cost before and after the policy update($J_c(\bar{\psi}_i) - J_c(\psi_i)$), is bounded by $L_{c,\psi_i}(\bar{\phi}_i)$. The proof is provided in Appendix C.5.

From Equation (22), we derive a lower bound on the expected policy improvement for agent $i$, which guarantees monotonic improvement under policy updates. From Equation (25), we derive an upper bound on the expected policy improvement for agent $i$, which constrains adaptive communication range selection. By jointly maximizing the lower bound in Equation (22) and minimizing the upper bound in Equation (25), we arrive at the following theorem. For notational convenience, let $L_{\psi_i}(\hat{\psi}_i) = L_{\psi_i}(\hat{\phi}_i) + L_{\psi_i}(\bar{\phi}_i, \hat{\pi}_i)$:

**Theorem 1** *Let $\bar{\psi}_i = \{\bar{\phi}_i, \bar{\pi}_i\}$. When the updated policy $\bar{\psi}_i$ satisfies both the monotonic improvement condition $J(\bar{\psi}_i) \geq J(\psi_i)$ and the communication cost constraint $J_c(\bar{\psi}_i) \leq c_i$, the policy update for agent $i$ is given by:*

$$\bar{\psi}_i = \arg\max_{\bar{\psi}_i}(L_{\psi_i}(\hat{\psi}_i) - M_{KL} - \mathbb{E}_{\kappa_{i,t} \sim \bar{\phi}_i(\cdot | s_{\mathcal{N}_{i,t}^{\kappa_{i,t}}})}[\eta' \rho^{\kappa_{i,t}+l}]),$$

$$\textbf{\textit{s.t}} \ J_c(\psi_i) + L_{c,\psi_i}(\bar{\phi}_i) - \nu_{c,\phi_i} D_{KL}^{\max}(\phi_i, \bar{\phi}_i) \leq c_i, \qquad M_{KL} \leq \delta, \tag{26}$$

*where* $M_{KL} = \nu_{\phi_i} D_{KL}^{\max}(\phi_i, \bar{\phi}_i) + \nu_{\pi_i} D_{KL}^{\max}(\pi_i, \bar{\pi}_i)$, $\eta' = \frac{2\bar{r} + (1-\gamma)G}{1-\gamma}$, $\sigma = \gamma$,

$$\nu_{c,\phi_i} = \frac{4\gamma \max_{s,\kappa_{i,t}} |A_{c,\psi_i}^{\bar{\phi}_i}(s_{\mathcal{N}_{i,t}^{\kappa_{i,t-1}}}, \kappa_{i,t})|}{(1-\gamma)^2}, \nu_{\phi_i} = \frac{4\gamma \max_{s,\kappa_{i,t}} |A_{\psi_i}^{\phi_i}(s_{\mathcal{N}_{i,t}^{\kappa_{i,t-1}}}, \kappa_{i,t})|}{(1-\gamma)^2},$$

$$\nu_{\pi_i} = \frac{4\gamma \max_{s,\kappa_{i,t}a_{i,t}} |\widetilde{A}_{\psi_i}^{\phi_i,\pi_i}(s_{\mathcal{N}_{i,t}^{\kappa_{i,t-1}}}, \kappa_{i,t}, a_{i,t}, a_{\mathcal{N}_{-i,t}^{\kappa_{i,t}}})|}{(1-\gamma)^2}, \mu_1 = J_c(\psi_i) + L_{c,\psi_i}(\bar{\phi}_i) - c_i,$$

$$\delta = \frac{\nu_{c,\phi_i}\mu_1 + \nu_{\phi_i}\nu_{\pi_i} D_{KL}^{max}(\pi_i, \bar{\pi}_i)}{\nu_{\phi_i}}.$$

Theorem 1 guarantees that agent performance can be monotonically improved while keeping the communication range within constraints. The proof is provided in Appendix C.6.

### 3.3 ALGORITHMS

We focus on the practical implementation of the policy update in Theorem 1, as outlined in Algorithm 1. Let $\theta_i$ denote the parameters of policy $\psi_i$, with $\theta_i^\phi$ and $\theta_i^\pi$ corresponding to the communication policy $\phi_i$ and the behavior policy $\pi_i$, respectively. For notational convenience, let

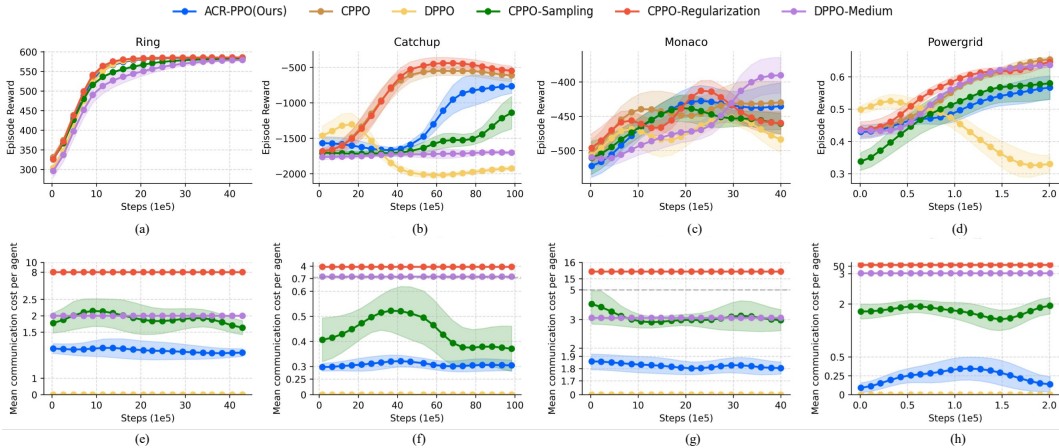

Figure 1: Policy performance and communication cost. (a)-(d) Policy performance of different approaches in Ring, Catchup, Monaco and Powergrid environments. (e)-(h) Average individual communication cost of different approaches in the corresponding environments.

$d_i = J_c(\bar{\psi}_i) - c_i$. $\lambda_i$ is a scalar variable. We apply Lagrangian relaxation to the communication cost constraint, yielding:

$$\max_{\theta_i^\phi, \theta_i^\pi} \min_{\lambda_i > 0} \mathbb{E}_{s, \kappa_{i,t}, a_{i,t}} \left[ \min \left( \frac{\bar{\phi}_i(\theta_i^\phi) \cdot \bar{\pi}_i(\theta_i^\pi)}{\phi_i(\theta_i^\phi) \cdot \pi_i(\theta_i^\pi)}, (1 \pm \epsilon) \frac{\bar{\phi}_i(\theta_i^\phi) \cdot \bar{\pi}_i(\theta_i^\pi)}{\phi_i(\theta_i^\phi) \cdot \pi_i(\theta_i^\pi)} \right) A_{\theta_i}^{\lambda_i, \theta_i^\phi, \theta_i^\pi} \right],$$

$$A_{\theta_i}^{\lambda_i, \theta_i^\phi, \theta_i^\pi} = \widetilde{A}_{\theta_i}^{\theta_i^\phi, \theta_i^\pi} \left( s_{\mathcal{N}_{i,t}^{\kappa_{i,t-1}}}, \kappa_{i,t}, a_{i,t}, a_{\mathcal{N}_{-i,t}^{\kappa_{i,t}}} \right) + A_{\theta_i}^{\theta_i^\phi} \left( s_{\mathcal{N}_{i,t}^{\kappa_{i,t-1}}}, \kappa_{i,t} \right)$$
$$- \lambda_i \left( A_{c,\theta_i}^{\theta_i^\phi} \left( s_{\mathcal{N}_{i,t}^{\kappa_{i,t-1}}}, \kappa_{i,t} \right) + d_i \right). \tag{27}$$

For more details on the Lagrangian derivation, see Appendix C.7. Each agent $i$ updates $\theta_i^\phi$ and $\theta_i^\pi$ by maximizing the surrogate objective in Equation (67), allowing the agent to simultaneously maximize the expected return while minimizing its communication range.

## 4 EXPERIMENTS

Our experiments aim to answer the following two questions:

- How effective is our approach in achieving high performance at low cost? (Subsection 4.2)
- Can the communication policy take effect? If so, how does it work? (Subsection 4.3)

### 4.1 EXPERIMENTS SETUP

**Environments.** We evaluate our approach on four representative networked multi-agent scenarios: (1) Cooperative Adaptive Cruise Control (CACC) (Chu et al., 2020), where the objective is to adaptively coordinate a fleet of 8 vehicles to minimize spacing and speed disturbances based on real-time V2V communication. This includes two scenarios: Catchup (16 Agents) and Slowdown (8 Agents). (2) Autonomous Vehicle Control (Flow) (Vinitsky et al., 2018), which shares similar objectives with CACC—maintaining target speed and avoiding collisions. It includes the Eight (14 Agents) and Ring (22 Agents). (3) Adaptive Traffic Signal Control (ATSC) (Chu et al., 2020), where the objective is to dynamically adjust traffic signal phases based on real-time traffic measurements to reduce congestion. It includes Grid (25 Agents), Monaco (28 Agents) and New York(436 Agents). (4) IEEE Power Grid (Chen et al., 2021), where the objective is to improve voltage control performance in an islanded microgrid system (20 Agents).

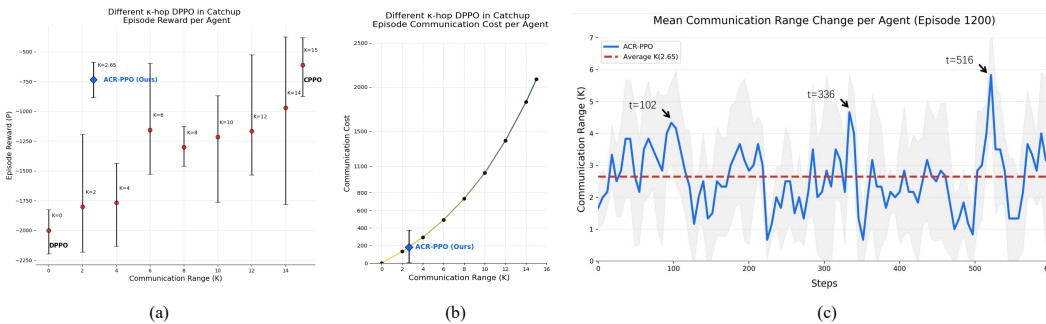

Figure 2: Communication policy analysis. (a) Different fixed $\kappa$-hop DPPO episode reward per agent in Catchup. (b) Different fixed $\kappa$-hop DPPO episode reward per agent in Catchup. (c) Mean Communication range change per agent.

**Baseline.** we evaluate the following algorithms in our experiments: (1) CPPO: The centralized PPO with a centralized critic trained using global information. This baseline evaluates agent performance when $\kappa$ is set to its maximum. (2) DPPO-Medium: This baseline is used to simulate the scenario where agents adopt a fixed, moderate communication range. (3) DPPO: The decentralized PPO learns with independent actor and critic using only local observations. This baseline analyzes agent performance when $\kappa$ is minimized. (4) CPPO-Sampling: We apply random masking to the input features to simulate random sampling. (5) CPPO-Regularization: We added L1 regularization to the agent input. All experimental results are based on 5 different random seeds. More environments and baseline's detail can be found in the Appendix D.1 and D.2.

### 4.2 POLICY PERFORMANCE AND COMMUNICATION COST CONTROL

Figure 1(a)-(d) show the policy performance of different approaches, with the horizontal axis representing steps and the vertical axis representing episode reward. Figure 1(e)-(h) show the average communication cost of different algorithms per agent in the corresponding scenarios. The communication cost is primarily based on (Er Meng, 2010), assuming that the cost increases exponentially with the expansion of the communication range. Since CPPO and DPPO use a fixed communication range, their corresponding communication costs remain constant.

**Experimental results demonstrate that**: (a) Since our approach focuses on adjusting the communication range, its performance is typically between CPPO and DPPO. By introducing a communication policy, our approach significantly reduces the communication cost. (b) In the Ring environment, all three approaches exhibit almost identical performance, as this environment is less sensitive to communication range, resulting in overlapping performance curves. (c) In the Catchup, Monaco, and PowerGrid environments, DPPO approach shows unstable policy performance due to non-stationarity issues, while our approach stabilizes learning through dynamic communication range adaptation. (d) The communication cost typically exhibits a "rise first, then decline" phenomenon. This suggests that, in the early training stages, agents increase the average communication range to ensure performance; once performance stabilizes, they subsequently reduce the average communication range to minimize cost. (e) CPPO-Regularization achieves slightly better performance than CPPO in some experiments, suggesting that the regularization term effectively suppresses redundant information. However, it does not reduce communication at the policy level. This is primarily because the L1 penalty only encourages the sum of the input vector to approach zero—it does not drive individual entries to exact sparsity, and thus cannot fully eliminate communication. (f) CPPO-Sampling achieves decent task performance but poorly controls communication cost. The main reason is that random sampling operates independently of the current state and cannot dynamically decide—based on situational demands—whether to adjust or suppress communication. (g) DPPO-Medium adopts a reduced communication range ($\kappa/2$) by halving the original value. Experimental results show that it still achieves certain performance, but due to an exponential increase in communication cost, the overall communication cost remains high.

**Significance**. Adding a communication policy incurs some training-time overhead; however, since this network is small, the training time of ACR-PPO does not increase significantly compared to

DPPO, which operates without communication but theoretically lacks convergence guarantees and performance advantages. Additionally, since we do not simulate the actual communication process between agents or between agents and a central server, CPPO achieves the shortest runtime. However, in real-world scenarios, the demand for extensive communication, along with issues such as time delay and packet loss caused by dense communication, imposes significant pressure on centralized training. Our work aims to mitigate these substantial overheads by learning communication policies, which is often a worthwhile endeavor. Details on the trainingtime and the design of network parameters are provided in the Appendix D.3.

Additional experimental results are provided in Appendix D.4 and D.5.

### 4.3 COMMUNICATION POLICY ANALYSIS

**Comparing with fixed $\kappa$-hop DPPO.** Figure 2(a) compares the performance of our approach with those of DPPO under fixed communication ranges, given the same number of training episodes. We use different fix $\kappa$ values: $\{0, 2, 4, 6, 8, 10, 12, 14, 15\}$. Figure 2(b) illustrates the average communication cost per agent over a single episode under different communication range $\kappa$. The experimental results show that: (a) Overall, as the communication range increases, the performance of DPPO improves; however, the associated communication cost grows exponentially. (b) Although agents achieve the best performance with global communication ($\kappa = 15$), this comes with significantly higher communication cost. In contrast, our approach adaptively adjusts the communication range based on the current communication state, significantly reducing communication cost. While achieving nearly the same performance as the full-communication setting, our approach reduces the average communication range to only $\kappa = 2.65$ per episode.

**Communication range change.** In order to further reveal the working mechanism of our approach, Figure 2(c) presents the evolution of the average communication range $\kappa$ per agent over step during the last training episode (episode=1200). The red line denotes the average $\kappa$ value under the ACR-PPO, as illustrated in Figure 2(a). The results indicate that agents do not require large-scale communication at every timestep. Maintaining a communication range of approximately $\kappa = 2$ to $4$ for most of the episode proves sufficient, and temporarily expanding $\kappa$ at critical timesteps ($t = 102, 336, 516$) can effectively achieve performance comparable to that of continuous large-scale communication. Further analysis on communication range variation can be found in the Appendix D.6.

## 5 RELATED WORKS

### 5.1 DECENTRALIZED MARL

As the number of agents increases, existing MARL approaches often face scalability challenges. A widely adopted strategy is decentralized MARL. Broadly, decentralized learning approaches can be categorized into two types: The first type pursues fully decentralized training, where each agent learns independently without any communication during training (De Witt et al., 2020; Sun et al., 2023; Su & Lu, 2024; Li et al., 2023b). Although these approach have achieved better performance, even matching centralized approaches, it relies on strong assumptions such as knowledge of others' policies or ideal conditions. The second type relaxes full decentralization by allowing agents to communication with local neighbors, called Networked MARL. Scalable AC (Qu et al., 2020; Zhang et al., 2023) incorporated spatial correlation decay into the learning process, while DMPO (Ma et al., 2024) combined this principle with model-based approaches to enhance sample efficiency. In additions, some works in safe MARL (Lu et al., 2021; Ying et al., 2023; Zhang et al., 2024b) also operate under this setting. These studies have significantly advanced the frontier of scalable MARL research. However, most existing approaches require manually setting the communication range $\kappa$-hop, which limits their adaptability across varying environments. In contrast, our approach follows the same assumptions but uses an adaptive communication mechanism, offering greater flexibility and scalability.

## 5.2 MARL WITH COMMUNICATION CONSTRAINT

More communication among agents can improve coordination, but it also increases the communication cost (Zhu et al., 2024). Some MARL approaches, including role-based (Sheng et al., 2022) and gating-based methods (Mao et al., 2020; Wang et al., 2023), have explored ways to reduce communication cost while maintaining performance. These approaches typically rely on global state information or the joint action-value function $Q_{tot}(s, a)$ to regulate communication, making them generally unsuitable for decentralized settings. Sampling-based methods Lin et al. (2021); Agorio et al. (2025) may offer theoretical guarantees, but they rely on a fixed graph distribution and fail when communication demands shift. Moreover, these sampling methods usually make communication stay around the average of the minimum and maximum communication range.

For a review of related work on sequential update, see Appendix E.1.

## 6 CONCLUSION

We propose a networked MARL method, called ACR-PPO. The key is that it models communication-aware decision-making under communication budget constraints as a sequential process. More importantly, our approach provides a theoretical guarantee of monotonic performance improvement under communication budget constraints. Experiments across diverse cooperative tasks demonstrate that ACR-PPO maintains policy performance while significantly reducing communication cost through adaptive range control. For future work, we will consider incorporating continual learning into ACR-PPO to enable adaptation to more diverse and dynamic environments.

## ETHICS STATEMENT

This work focuses on developing reinforcement learning algorithms and does not involve human subjects, personal data, or sensitive information. The experiments are conducted on publicly available benchmark datasets and simulated environments. We believe our research raises no direct ethical concerns and may contribute positively by improving the scalable cooperative MARL methods.

## REPRODUCIBILITY STATEMENT

All implementation details, including source code, hyperparameters, and scripts, are provided in the appendix and supplementary material to enable full reproducibility of our results.

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

## A  PRELIMINARIES LEMMA

Before proving the main lemmas, we first establish a series of intermediate results.

**Lemma 3** *For any agent $i \in \mathcal{N}$, let $s = (s_{\mathcal{N}_i^\kappa}, s_{\mathcal{N}_{-i}^\kappa})$ and $s' = (s_{\mathcal{N}_i^\kappa}, s'_{\mathcal{N}_{-i}^\kappa})$ denote two states that differ only in the states of agents outside the communication neighborhood of agent $i$. Let $\boldsymbol{\mu}, \boldsymbol{\nu}$ be two joint distributions over the state space $\mathcal{S}$, and let $\mu^i, \nu^i$ be the corresponding marginal distributions over $\mathcal{S}^i$ for agent $i$. Define $\delta^i(f) = \sup_{s^i, s^{-i}, s'^{-i}} \left| f(s^i, s^{-i}) - f(s^i, s'^{-i}) \right|$, which measures the maximum variation of function $f(s)$ with respect to changes in the states of agents other than $i$, while keeping agent $i$'s state fixed. Let the L1 norm be defined as $\|f\|_1 = \int |f(s)| \, ds$. Then,*

$$|\mathbb{E}_{s\sim\boldsymbol{\mu}}[f(s)] - \mathbb{E}_{s\sim\boldsymbol{\nu}}[f(s)]| \leq \sum_{i\in\mathcal{N}} D_{\text{TV}}(\mu^i, \nu^i) \cdot \delta^i(f). \tag{28}$$

**Proof**. We first present the mathematical expression for the total variation distance $D_{TV}$:

$$D_{TV}(\boldsymbol{\mu}, \boldsymbol{\nu}) = \frac{1}{2} \sup_h |\mathbb{E}_{\boldsymbol{\mu}}(h) - \mathbb{E}_{\boldsymbol{\nu}}(h)|$$

$$= \frac{1}{2} \int |\boldsymbol{\mu}(s) - \boldsymbol{\nu}(s)| ds. \tag{29}$$

In multi-agent systems, the number of agents is typically greater than one, i.e., $N = |\mathcal{N}| \geq 2$. Based on this, we can obtain:

$$
\begin{aligned}
|\mathbb{E}_{\boldsymbol{\mu}(s)} f(s) - \mathbb{E}_{\boldsymbol{\nu}(s)} f(s)| &= |\int f(s)\boldsymbol{\mu}(s)ds - \int f(s)\boldsymbol{\nu}(s)ds| \\
&= \int |f(s)[\boldsymbol{\mu}(s) - \boldsymbol{\nu}(s)]|ds \\
&\leq \int |f(s)||\boldsymbol{\mu}(s) - \boldsymbol{\nu}(s)|ds \\
&\leq ||f(s)||_1 \cdot \int |\boldsymbol{\mu}(s) - \boldsymbol{\nu}(s)|ds \qquad (30) \\
&= ||f(s)||_1 \cdot 2D_{TV}(\boldsymbol{\mu}, \boldsymbol{\nu}) \\
&\leq \sum_{i=1}^{\mathcal{N}} D_{TV}(\mu^i, \nu^i) \cdot \delta^i(f(s)). \qquad (31)
\end{aligned}
$$

The proof is complete. Equation (3) follows from Hölder's inequality. (Young, 1936).

# B PRELIMINARIES

## B.1 SPATIAL CORRELATION DECAY

Our approach primarily relies on the spatial exponential decay assumption, which states that the influence between agents decays exponentially with their distance in the communication graph. This assumption is well-established and widely used across various domains in multi-agent systems. Over the past decades, many researchers have leveraged the spatial correlation property, to design scalable, distributed algorithms for optimization and control in multi-agent settings. For example: In power networks, the concept of spatial exponential decay has been employed to design decentralized secondary voltage control schemes for inverters in distributed generation systems (Chen et al., 2021). In epidemiological dynamics, the existence of critical traveling wave solutions exhibiting exponential decay has been established in K-M epidemic models (Deng et al., 2022). In traffic flow networks, the temporal variation of distance-decay effects modeled through spatial exponential functions has been shown to play a key role in shaping the evolution of inter-city spatial interactions (Li et al., 2020).

Based on spatial correlation decay, we can formulate the exponential decay assumption:

**Assumption 1** *(Spatial Decay of Correlation for the Dynamics) Assume that there exist $\beta > 0$, for any agents $i, j \in \mathcal{N}$, such that*

$$
max_{i \in \mathcal{N}} \sum_{j \in \mathcal{N}} e^{\beta d(i,j)} W^{ij} \leq \zeta, \qquad (32)
$$

*where $d(i,j)$ represents the distance between agent $i$ and agent $j$, and $\zeta \in [0, 2/\gamma)$ is a constant. $W^{ij} = \sup \|P^i(\cdot|z^j, z^{-j}) - P^i(\cdot|z^j, z^{-j})\|_1$, $z_j = (s_j, a_j)$ and $z'_j = (s'_j, a'_j)$ represent two different state-action pairs of the agent $j$ respectively, and $z'_j$ represents the state-action pair of the agent other than $j$. The value of $W^{ij}$ reflects the extent to which the local transition probability of agent $i$ is affected by the state and action of agent $j$.*

Assumption 1 portrays a common phenomenon: the transition dynamic of each agent is exponentially less sensitive to perturbations of the states and actions of more distant agents.

**Assumption 2** *(Spatial Decay of Correlation for the Policies) Assume that there exist $\xi, \beta \geq 0$ such that for any agent $i \in \mathcal{N}$, $\mathrm{s}_{\mathcal{N}_\kappa^i} \in \mathcal{S}_{\mathcal{N}_\kappa^i}, \mathbf{s}_{\mathcal{N}_\kappa^{-i}}, \mathbf{s}'_{\mathcal{N}_\kappa^{-i}} \in \mathcal{S}_{\mathcal{N}_\kappa^{-i}}$, one have*

$$
\sup_{\mathrm{s}_{\mathcal{N}_\kappa^i}, \mathbf{s}_{\mathcal{N}_\kappa^{-i}}, \mathbf{s}'_{\mathcal{N}_\kappa^{-i}}} \left| \pi^i\left(\cdot | \mathrm{s}_{\mathcal{N}_\kappa^i}, \mathbf{s}_{\mathcal{N}_\kappa^{-i}}\right) - \pi^i\left(\cdot | \mathrm{s}_{\mathcal{N}_\kappa^i}, \mathbf{s}'_{\mathcal{N}_\kappa^{-i}}\right) \right| \leq \xi e^{-\beta \kappa}. \qquad (33)
$$

Assumption 2 reveals how much information is lost compared with access to the global state and allows us to consider a policy class with the necessary properties for the optimal policy under Assumption 2.

### B.2 THE PROOF OF LEMMA 1

For notational simplicity, we define $s = (s_{\mathcal{N}_i^\kappa}, s_{\mathcal{N}_{-i}^\kappa})$, $a = (a_{\mathcal{N}_i^\kappa}, a_{\mathcal{N}_{-i}^\kappa})$, $s' = (s_{\mathcal{N}_i^\kappa}, s'_{\mathcal{N}_{-i}^\kappa})$, $a' = (a_{\mathcal{N}_i^\kappa}, a'_{\mathcal{N}_{-i}^\kappa})$. Let $\rho_{i,t}$ denote the state-action visitation distribution at timestep $t$ under policy $\psi$ starting from $(s(0), a(0)) = (s, a)$, and let $\rho'_{i,t}$ denote the corresponding distribution starting from $(s(0), a(0)) = (s', a')$. We require $\rho_{i,t} = \rho'_{i,t}$ for all $t \leq \kappa$. The reason is that, due to the local dependence structure in Equation (1) and the spatial decay assumptions (Assumption 1 and Assumption 2), if the range of neighborhood dependence changes in Equation (1), "the propagation rate per unit time" must be adjusted accordingly. For convenience in analysis, it is typically set to 1. With these definitions, we now introduce the truncated $Q$-function:

$$|Q_i^\theta(s, a) - Q_i^\theta(s', a')|$$

$$= \sum_{t=0}^{\infty} |\gamma^t \mathbb{E}_{(s_i, a_i) \sim \rho_{i,t}} r_i(s_i, a_i) - \gamma^t \mathbb{E}_{(s_i, a_i) \sim \rho'_{i,t}} r_i(s'_i, a'_i)|$$

$$= \sum_{t=0}^{\kappa} |\gamma^t \mathbb{E}_{(s_i, a_i) \sim \rho_{i,t}} r_i(s_i, a_i) - \gamma^t \mathbb{E}_{(s_i, a_i) \sim \rho'_{i,t}} r_i(s_i, a_i)|$$

$$+ \sum_{t=\kappa+1}^{\infty} |\gamma^t \mathbb{E}_{(s_i, a_i) \sim \rho_{i,t}} r_i(s_i, a_i) - \gamma^t \mathbb{E}_{(s_i, a_i) \sim \rho'_{i,t}} r_i(s_i, a_i)| \tag{34}$$

$$= \sum_{t=\kappa+1}^{\infty} \gamma^t |\mathbb{E}_{(s_i, a_i) \sim \rho_{i,t}} r_i(s_i, a_i) - \mathbb{E}_{(s_i, a_i) \sim \rho'_{i,t}} r_i(s_i, a_i)|$$

$$\leq \sum_{t=\kappa+1}^{\infty} \gamma^t \bar{r} TV(\rho_{i,t}, \rho'_{i,t}) \leq \frac{\bar{r}}{1-\gamma} \gamma^{\kappa+1}. \tag{35}$$

In Equation (34), the first term equals 0 because $\rho_{i,t} = \rho'_{i,t}$ for all $t \leq \kappa$. The derivation of the final step is provided in Appendix 3. Equation (35) holds because the TV distance is bounded by 1. Detailed descriptions of this part can be found in (Qu et al., 2020).

Next, we show that the truncated $Q$-function is a good approximation of the global $Q$-function. Let $(c = \frac{\bar{r}}{1-\gamma}, \rho = \gamma)$. From the above bound, it follows that:

$$|\widetilde{Q}_i^\theta(s_{\mathcal{N}_i^\kappa}, a_{\mathcal{N}_i^\kappa}) - Q_i^\theta(s, a)|$$

$$= \Big| \sum_{s'_{\mathcal{N}_{-i}^\kappa}, a'_{\mathcal{N}_{-i}^\kappa}} \omega_i(s'_{\mathcal{N}_i^\kappa}, a'_{\mathcal{N}_{-i}^\kappa}; s_{\mathcal{N}_i^\kappa}, a_{\mathcal{N}_{-i}^\kappa}) Q_i^\theta(s_{\mathcal{N}_i^\kappa}, s'_{\mathcal{N}_{-i}^\kappa}, a_{\mathcal{N}_i^\kappa}, a'_{\mathcal{N}_{-i}^\kappa}) - Q_i^\theta(s_{\mathcal{N}_i^\kappa}, s_{\mathcal{N}_{-i}^\kappa}, a_{\mathcal{N}_i^\kappa}, a_{\mathcal{N}_{-i}^\kappa})\Big|$$

$$\leq \sum_{s'_{\mathcal{N}_{-i}^\kappa}, a'_{\mathcal{N}_{-i}^\kappa}} \omega_i(s'_{\mathcal{N}_i^\kappa}, a'_{\mathcal{N}_i^\kappa}; s_{\mathcal{N}_i^\kappa}, a_{\mathcal{N}_{-i}^\kappa}) |Q_i^\theta(s_{\mathcal{N}_i^\kappa}, s'_{\mathcal{N}_{-i}^\kappa}, a_{\mathcal{N}_i^\kappa}, a'_{\mathcal{N}_{-i}^\kappa}) - Q_i^\theta(s_{\mathcal{N}_i^\kappa}, s_{\mathcal{N}_{-i}^\kappa}, a_{\mathcal{N}_i^\kappa}, a_{\mathcal{N}_{-i}^\kappa})|$$

$$\leq c\rho^{\kappa+1}. \tag{36}$$

Similarly, the value function $V$ exhibits a spatial correlation decay:

$$|V_i^\theta(s) - V_i^\theta(s')| \leq \frac{\bar{r}}{1-\gamma} \gamma^{\kappa_{i,t}+1}, \tag{37}$$

$$|\widetilde{V}_i^\theta(s_{\mathcal{N}_i^\kappa}) - V_i^\theta(s)| \leq c\rho^{\kappa+1}. \tag{38}$$

From this, we derive the following expression for $\hat{h}_i - \nabla_{\theta_i} J(\theta)$:

$$\hat{g}_i(\theta) - \nabla_{\theta_i} J(\theta) = \mathbb{E}_{a_i \sim \pi_i(\cdot|s_i)}[\nabla_{\theta_i} \log \pi_i(a_i|s_i)(\frac{1}{n} \sum_{j \in \mathcal{N}_i^{\kappa}} \widetilde{Q}_j^{\theta}(s_{\mathcal{N}_i^{\kappa}}, a_{\mathcal{N}_i^{\kappa}}) - Q(\mathbf{s}, \mathbf{a}))]$$

$$= \mathbb{E}_{a_i \sim \pi_i(\cdot|s_i)}[\nabla_{\theta_i} \log \pi_i(a_i|s_i)(\frac{1}{n} \sum_{j \in \mathcal{N}_i^{\kappa}} \widetilde{Q}_j^{\theta}(s_{\mathcal{N}_i^{\kappa}}, a_{\mathcal{N}_i^{\kappa}}) - \frac{1}{n} \sum_{j \in \mathcal{N}} Q_j(s_j, a_j))]$$

$$- \mathbb{E}_{a_i \sim \pi_i(\cdot|s_i)}[\nabla_{\theta_i} \log \pi_i(a_i|s_i)(\frac{1}{n} \sum_{j \in \mathcal{N}_{-i}^{\kappa}} \widetilde{Q}_j^{\theta}(s_{\mathcal{N}_j^{\kappa}}, a_{\mathcal{N}_j^{\kappa}}))]$$

$$:= E_1 - E_2. \tag{39}$$

Now we prove that $E_2 = 0$. Recall that the joint policy is given by $\pi(s, a) = \prod_{l=1}^{n} d(s) \pi(a_l|s_l)$, where $d(s)$ denotes the stationary state distribution. Then, for any $j \in \mathcal{N}_{-i}^{\kappa}$, we have:

$$E_2 = \mathbb{E}_{a_i \sim \pi_i(\cdot|s_i)}[\nabla_{\theta_i} \log \pi_i(a_i|s_i) Q_j(s_j, a_j)]$$

$$= \sum_{s,a} d(s) \prod_{l=1}^{n} \pi(a_l|s_l) \frac{\nabla_{\theta_i} \pi_i(a_i|s_i)}{\pi_i(a_i|s_i)} \widetilde{Q}_j^{\theta}(s_{\mathcal{N}_j^{\kappa}}, a_{\mathcal{N}_j^{\kappa}})$$

$$= 0. \tag{40}$$

We can bound $E_1$ as follows:

$$||\hat{g}_i(\theta) - \nabla_{\theta_i}(\theta)|| = ||E_1|| \leq \mathbb{E}_{a_i \sim \pi_i(\cdot|s_i)}||\nabla_{\theta_i} \log \pi_i(a_i|s_i)|| \frac{1}{n} \sum_{j \in \mathcal{N}} |\widetilde{Q}_j^{\theta}(s_{\mathcal{N}_i^{\kappa}}, a_{\mathcal{N}_i^{\kappa}}) - Q_j(s_j, a_j))|$$

$$\leq L_i c \rho^{\kappa+1}. \tag{41}$$

This completes the proof.

### B.3 COMMUNICATION COST

In networked cooperative multi-agent systems, multi-hop communication enables the connection of more agents. However, the communication cost also increases exponentially. In this paper, we follow the setting of (Er Meng, 2010) and model the communication cost as an exponentially increasing function of the communication range:

$$c_{i,t} = e^{\alpha \kappa_{i,t}} - 1, \tag{42}$$

where $\alpha$ is a hyperparameter, setting to 0.1.

### B.4 JOINT STATE-ACTION VALUE FUNCTION

The detail of the joint state-action value function is as follows:

$$Q(\mathbf{s}, \mathbf{a})$$

$$= \mathbb{E}_{\mathbf{a} \sim \psi}[\sum_{t=0}^{\infty} \gamma^t r(\mathbf{s}(t), \mathbf{a}(t))|\mathbf{s}(0) = \mathbf{s}, \mathbf{a}(0) = \mathbf{a}]$$

$$= \frac{1}{n} \sum_{i=1}^{n} \mathbb{E}_{\mathbf{a} \sim \psi}[\sum_{t=0}^{\infty} \gamma^t r_i(s_i(t), a_i(t))|\mathbf{s}(0) = \mathbf{s}, \mathbf{a}(0) = \mathbf{a}]$$

$$:= \frac{1}{n} \sum_{i=1}^{n} Q_i^{\theta}(\mathbf{s}, \mathbf{a}). \tag{43}$$

### B.5 MULTI-AGENT POLICY GRADIENT

**Lemma 4** *Let $\nu^{\theta}$ be a distribution over the state space defined as $\nu^{\theta}(\mathbf{s}) = (1 - \gamma) \sum_{t=0}^{\infty} \gamma^t \nu_t^{\theta}(\mathbf{s})$, where $\pi^{\theta}(\mathbf{s})$ represents the distribution of $\mathbf{s}(t)$ under a fixed joint policy parameter $\theta = (\theta_1, \theta_2, ..., \theta_n)$, given that $\mathbf{s}(0)$ is sampled from $\pi_0$. $\psi$ is the joint policy. Then we have,*

$$\nabla J(\theta) = \frac{1}{1 - \gamma} \mathbb{E}_{\mathbf{s} \sim \nu^{\theta}, \mathbf{a} \sim \psi(\cdot|\mathbf{s}(t))} \left[ \nabla \log \psi(\mathbf{a}|\mathbf{s}) Q^{\theta}(\mathbf{s}, \mathbf{a}) \right]. \tag{44}$$

### B.6 MAPPO LOSS FUNCTION

The loss function of MAPPO (Yu et al., 2022) is defined as:

$$L_{actor}(\theta_k) = J(\theta_k) - J(\theta_k^{old}) = \frac{1}{B}\sum_{i=1}^{B}\sum_{k=1}^{n}\min\left[U, clip\left(U, 1 \pm \epsilon\right)\right]A(\mathbf{s},\mathbf{a}),$$

$$U = \frac{\psi_{\theta_i}(a_k|o_k)}{\psi_{\theta_k^{old}}(a_k|o_k)}. \tag{45}$$

$$L_{critic}(\theta_k) = \frac{1}{B}\sum_{i=1}^{B}\sum_{k=1}^{n}\left(\max\left[(V-R)^2, (clip(V, V \pm \epsilon) - R)^2\right]\right). \tag{46}$$

The advantage function is defined as $A(\mathbf{s},\mathbf{a}) = r_t(\mathbf{s},\mathbf{a}) + V(\mathbf{s}_{t+1}) - V(\mathbf{s}_t)$, with $B$ denoting the batch size. Agents share a common actor and critic network to enhance training efficiency.

## C METHODS

### C.1 THE PROOF OF PROPOSITION 1

Let $s_{\mathcal{N}_{i,t}^{\kappa_{i,t}}} = (s_{\mathcal{N}_{i,t}^{\kappa_{i,t-1}}}, s_{i,t})$, $a_{\mathcal{N}_{i,t}^{\kappa_{i,t}}} = (a_{i,t}, a_{\mathcal{N}_{-i,t}^{\kappa_{i,t}}})$, and define the truncated joint state-action tuples as $\mathbf{s}_{\kappa_{i,t}} = (s_{\mathcal{N}_{i,t}^{\kappa_{i,t}}}, s_{\mathcal{N}_{-i,t}^{\kappa_{i,t}}})$, $\mathbf{a}_{\kappa_{i,t}} = (a_{\mathcal{N}_{i,t}^{\kappa_{i,t}}}, a_{\mathcal{N}_{-i,t}^{\kappa_{i,t}}})$, $\mathbf{s}'_{\kappa_{i,t}} = (s_{\mathcal{N}_{i,t}^{\kappa_{i,t}}}, s'_{\mathcal{N}_{-i,t}^{\kappa_{i,t}}})$, $\mathbf{a}'_{\kappa_{i,t}} = (a_{\mathcal{N}_{i,t}^{\kappa_{i,t}}}, a'_{\mathcal{N}_{-i,t}^{\kappa_{i,t}}})$. Let $\rho_{i,t}$ denote the state-action visitation distribution at timestep $t$ under policy $\psi_i$ starting from $(\mathbf{s}_{\kappa_{i,t}}, \mathbf{a}_{\kappa_{i,t}})$, and let $\rho'_{i,t}$ denote the corresponding distribution starting from $(\mathbf{s}'_{\kappa_{i,t}}, \mathbf{a}'_{\kappa_{i,t}})$. For a fixed constant $l$, we assume $\rho_{i,t} = \rho'_{i,t}$ for all $u \le l' + \kappa_{i,t}$. With these definitions, we now define the truncated $Q$-function at timestep $t$:

$$|Q_i^\theta(\mathbf{s},\mathbf{a}) - Q_i^\theta(\mathbf{s}',\mathbf{a}')|$$

$$= \mathbb{E}_{\kappa_{i,t}\sim\phi_i(\mathbf{s}(t))}\sum_{u=l'}^{\infty}|\gamma^u\mathbb{E}_{(s_i,a_i)\sim\rho_{i,t}}r_i(s_i,a_i) - \gamma^u\mathbb{E}_{(s_i,a_i)\sim\rho'_{i,t}}r_i(s'_i,a'_i)|$$

$$= \mathbb{E}_{\kappa_{i,t}\sim\phi_i(\mathbf{s}(t))}\sum_{u=l'+\kappa_{i,t}+1}^{\infty}|\gamma^u\mathbb{E}_{(s_i,a_i)\sim\rho_{u,i}}r_i(s_i,a_i) - \gamma^u\mathbb{E}_{(s_i,a_i)\sim\rho'_{u,i}}r_i(s_i,a_i)|$$

$$= \mathbb{E}_{\kappa_{i,t}\sim\phi_i(\mathbf{s}(t))}\sum_{u=l'+\kappa_{i,t}+1}^{\infty}\gamma^t|\mathbb{E}_{(s_i,a_i)\sim\rho_{u,i}}r_i(s_i,a_i) - \mathbb{E}_{(s_i,a_i)\sim\rho'_{u,i}}r_i(s_i,a_i)|$$

$$\le \mathbb{E}_{\kappa_{i,t}\sim\phi_i(\mathbf{s}(t))}\sum_{u=l'+\kappa_{i,t}+1}^{\infty}\gamma^u\bar{r}TV(\rho_{u,i},\rho'_{u,i}) \le \mathbb{E}_{\kappa_{i,t}\sim\phi_i(\mathbf{s}(t))}\frac{\bar{r}}{1-\gamma}\gamma^{l'+\kappa_{i,t}+1}. \tag{47}$$

Let $\eta = \frac{\bar{r}}{1-\gamma}, \sigma = \gamma$. For notational convenience, we shift the index by defining $l = l' + 1$, which absorbs the constant offset. Then, we obtain:

$$|Q_i^\theta(\mathbf{s},\mathbf{a}) - Q_i^\theta(\mathbf{s}',\mathbf{a}')| \le \mathbb{E}_{\kappa_{i,t}\sim\phi_i(\mathbf{s}(t))}\eta\sigma^{\kappa_{i,t}+l}. \tag{48}$$

This completes the proof.

### C.2 THE PROOF OF COROLLARY 1

Let $s_{\mathcal{N}_{i,t}^{\kappa_{i,t}}} = (s_{\mathcal{N}_{i,t}^{\kappa_{i,t-1}}}, s_{i,t})$, $a_{\mathcal{N}_{i,t}^{\kappa_{i,t}}} = (a_{i,t}, a_{\mathcal{N}_{-i,t}^{\kappa_{i,t}}})$, denote the state and action within agent $i$'s $\kappa_{i,t}$-hop neighborhood at timestep $t$. For notational simplicity, we drop the timestep index and define the truncated joint state-action tuples as $s_\kappa = (s_{\mathcal{N}_i^\kappa}, s_{\mathcal{N}_{-i}^\kappa})$, $a_\kappa = (a_{\mathcal{N}_i^\kappa}, a_{\mathcal{N}_{-i}^\kappa})$, $s'_\kappa = (s_{\mathcal{N}_i^\kappa}, s'_{\mathcal{N}_{-i}^\kappa})$, $a'_\kappa = (a_{\mathcal{N}_i^\kappa}, a'_{\mathcal{N}_{-i}^\kappa})$. Let $\rho_{t,i,\kappa}$ denote the state-action visitation distribution at timestep $t$ under policy $\psi$, starting from $(s(0), a(0)) = (s_\kappa, a_\kappa)$, and let $\rho'_{t,i,\kappa}$ be the corresponding

distribution starting from $(s(0), a(0)) = (s'_\kappa, a'_\kappa)$. We assume that $\rho_{t,i,\kappa} = \rho'_{t,i,\kappa}$ for all $t \leq \kappa$, meaning that the influence of distant agents outside $\mathcal{N}_i^\kappa$ vanishes within $\kappa$ steps. Then, for any agent $i \in \mathcal{N}$, and under the assumption that the states and actions of distant agents differ, we obtain the following bound:

$$
\begin{aligned}
|\widetilde{A}_{\psi_i}(s_{\mathcal{N}_\kappa^i}, a_{\mathcal{N}_\kappa^i}) - A_{\psi_i}(\mathbf{s}, \mathbf{a})| &= |(\widetilde{Q}_{\psi_i}(s_{\mathcal{N}_\kappa^i}, a_{\mathcal{N}_\kappa^i}) - \widetilde{V}_{\psi_i}(s_{\mathcal{N}_\kappa^i})) - (Q_{\psi_i}(\mathbf{s}, \mathbf{a}) - V_{\psi_i}(\mathbf{s}))| \\
&= |(\widetilde{Q}_{\psi_i}(s_{\mathcal{N}_\kappa^i}, a_{\mathcal{N}_\kappa^i}) - Q_{\psi_i}(\mathbf{s}, \mathbf{a})) + (V_{\psi_i}(\mathbf{s}) - \widetilde{V}_{\psi_i}(s_{\mathcal{N}_\kappa^i}))| \\
&\leq \underbrace{|\widetilde{Q}_{\psi_i}(s_{\mathcal{N}_\kappa^i}, a_{\mathcal{N}_\kappa^i}) - Q_{\psi_i}(\mathbf{s}, \mathbf{a})|}_{(1)} + \underbrace{|\widetilde{V}_{\psi_i}(s_{\mathcal{N}_\kappa^i}) - V_{\psi_i}(\mathbf{s})|}_{(2)}. \quad (49)
\end{aligned}
$$

We analyze these two terms separately. For term (1), the proof can be found in Appendix C.1. For term (2), we have:

$$
\begin{aligned}
&|\widetilde{V}_\psi(\mathbf{s}'_\kappa) - V_\psi(\mathbf{s}_\kappa)| \\
&= |\mathbb{E}_{\mathbf{a}'_\kappa \sim \psi(\cdot|\mathbf{s}'_\kappa)}\widetilde{Q}(\mathbf{s}'_\kappa, \mathbf{a}'_\kappa) - \mathbb{E}_{\mathbf{a}_\kappa \sim \psi(\cdot|\mathbf{s}_\kappa)}Q(\mathbf{s}_\kappa, \mathbf{a}_\kappa)| \\
&\leq |\mathbb{E}_{\mathbf{a}'_\kappa \sim \psi(\cdot|\mathbf{s}'_\kappa)}\widetilde{Q}(\mathbf{s}'_\kappa, \mathbf{a}'_\kappa) - \mathbb{E}_{\mathbf{a}'_\kappa \sim \psi(\cdot|\mathbf{s}'_\kappa)}Q(\mathbf{s}_\kappa, \mathbf{a}_\kappa)| + |\mathbb{E}_{\mathbf{a}'_\kappa \sim \psi(\cdot|\mathbf{s}'_\kappa)}Q(\mathbf{s}_\kappa, \mathbf{a}_\kappa) - \mathbb{E}_{\mathbf{a}_\kappa \sim \psi(\cdot|\mathbf{s}_\kappa)}Q(\mathbf{s}_\kappa, \mathbf{a}_\kappa)| \\
&= \frac{\bar{r}}{1-\gamma}\gamma^{\kappa_{i,t}+l} + |\mathbb{E}_{\mathbf{s}'_\kappa \sim \rho'_{t,i,\kappa}}\mathbb{E}_\psi Q(\mathbf{s}_\kappa, \mathbf{a}_\kappa) - \mathbb{E}_{\mathbf{s}_\kappa \sim \rho_{t,i,\kappa}}\mathbb{E}_\psi Q(\mathbf{s}_\kappa, \mathbf{a}_\kappa)| \\
&\leq \frac{\bar{r}}{1-\gamma}\gamma^{\kappa_{i,t}+l} + \sum_{i=1}^{\mathcal{N}} D_{TV}(\rho_{t,i,\kappa}||\rho'_{t,i,\kappa})\delta_i[\mathbb{E}_\psi Q(\mathbf{s}_\kappa, \mathbf{a}_\kappa)] \\
&\leq \frac{\bar{r}}{1-\gamma}\gamma^{\kappa_{i,t}+l} + \mathcal{N}\bar{r}. \quad (50)
\end{aligned}
$$

Here, $\bar{r}$ denotes the upper bound of the reward function. The term $\mathcal{N}\bar{r}$, treated as a constant, is bounded by $G\gamma^{\kappa+1}$ for some $G > 0$. To simplify the calculation, in this paper, we set $l'$ as the constant 1. Combining Equation equation 36, equation 49, and equation 50, we obtain:

$$
|\widetilde{A}_{\psi_i}(s_{\mathcal{N}_\kappa^i}, a_{\mathcal{N}_\kappa^i}) - A_{\psi_i}(\mathbf{s}, \mathbf{a})| \leq \frac{2\bar{r} + (1-\gamma)G}{1-\gamma}\gamma^{\kappa_{i,t}+l}. \quad (51)
$$

Let $\eta' = \frac{2\bar{r}+(1-\gamma)G}{1-\gamma}, \sigma = \gamma$. This completes the proof.

### C.3 THE PROOF OF COROLLARY 2

*Proof.* The surrogate objective for agent $i$ under policy $\psi_i = (\phi_i, \pi_i)$ is given by

$$
L_{\psi_i}(\phi_i, \pi_i) = \mathbb{E}_{s \sim \rho_i, \kappa_{i,t} \sim \phi_i(\cdot|s_{\mathcal{N}_{i,t}^{\kappa_i,t-1}}), a_{i,t} \sim \pi_i(\cdot|s_{\mathcal{N}_{i,t}^{\kappa_i,t-1}}, \kappa_{i,t})} \left[ A_{\psi_i}^{\phi_i, \pi_i}(s_{\mathcal{N}_{i,t}^{\kappa_i,t-1}}, \kappa_{i,t}, a_{i,t}, a_{\mathcal{N}_{-i,t}^{\kappa_i,t}}) \right], \quad (52)
$$

and The surrogate objective with global communication under policy $\psi_i$ is defined as

$$
L_{\psi_i}(\pi_i) = \mathbb{E}_{\mathbf{s} \sim \rho, a \sim \pi_i(\cdot|\mathbf{s})} \left[ A_{\psi_i}(\mathbf{s}, \mathbf{a}) \right]. \quad (53)
$$

By applying the reparameterization trick (Salimans & Kingma, 2016), we rewrite $L_{\psi_i}(\pi_i)$ in terms of an expectation over $\kappa_{i,t} \sim \phi_i^{full}(\cdot|s_{\mathcal{N}_{i,t}^{\kappa_i,t-1}})$:

$$
L_{\psi_i}(\phi_i^{full}, \pi_i) = \mathbb{E}_{s \sim \rho_i, \kappa_{i,t} \sim \phi_i^{full}(\cdot|s_{\mathcal{N}_{i,t}^{\kappa_i,t-1}}), a_{i,t} \sim \pi_i(\cdot|s_{\mathcal{N}_{i,t}^{\kappa_i,t-1}}, \kappa_{i,t})} A_{\psi_i}(s_\kappa, a_\kappa)]. \quad (54)
$$

Combining Equation (52), (54) and Lemma 3, we can obtain:

$$|L_{\psi_i}(\bar{\phi}_i, \bar{\pi}_i) - L_{\psi_i}(\phi_i^{full}, \bar{\pi}_i)|$$

$$\leq |\mathbb{E}_{s\sim\rho_i, \kappa_{i,t}\sim\phi_i(\cdot|s_{\mathcal{N}_{i,t}^{\kappa_{i,t-1}}}), a_{i,t}\sim\pi_i(\cdot|s_{\mathcal{N}_{i,t}^{\kappa_{i,t-1}}},\kappa_{i,t})}[A_{\psi_i}^{\phi_i,\pi_i}(s_{\mathcal{N}_{i,t}^{\kappa_{i,t-1}}}, \kappa_{i,t}, a_{i,t}, a_{\mathcal{N}_{-i,t}^{\kappa_{i,t}}})]$$

$$- \mathbb{E}_{s\sim\rho_i, \kappa_{i,t}\sim\phi_i^{full}(\cdot|s_{\mathcal{N}_{i,t}^{\kappa_{i,t-1}}}), a_{i,t}\sim\pi_i(\cdot|s_{\mathcal{N}_{i,t}^{\kappa_{i,t-1}}},\kappa_{i,t})}A_{\psi_i}(s_\kappa, a_\kappa)]|$$

$$= \mathbb{E}_{s\sim\rho_i, \kappa_{i,t}\sim\phi_i(\cdot|s_{\mathcal{N}_{i,t}^{\kappa_{i,t-1}}}), a_{i,t}\sim\pi_i(\cdot|s_{\mathcal{N}_{i,t}^{\kappa_{i,t-1}}},\kappa_{i,t})} \left| A_{\psi_i}^{\phi_i,\pi_i}(s_{\mathcal{N}_{i,t}^{\kappa_{i,t-1}}}, \kappa_{i,t}, a_{i,t}, a_{\mathcal{N}_{-i,t}^{\kappa_{i,t}}}) - A_{\psi_i}(s_\kappa, a_\kappa) \right|$$

$$\leq \frac{2\bar{r} + (1-\gamma)G}{1-\gamma} \mathbb{E}_{\kappa_{i,t}\sim\phi(s_t)}\gamma^{\kappa_{i,t}+l}. \tag{55}$$

Finally, denoting $(\eta' = \frac{2\bar{r}+(1-\gamma)G}{1-\gamma}, \sigma = \gamma)$, we can obtain Corollary 2.

## C.4 THE PROOF OF PROPOSITION 2

*Proof.* Equation (20) is derived from the TRPO framework (Schulman et al., 2015). Substituting Corollary 1 into the trust region method gives Equation (21). Combining Equation (20), (21) and applying the multi-agent trust region theorem from HATRPO (Kuba et al., 2022), it follows that:

$$J(\bar{\psi}_i) - J(\psi_i)$$

$$\geq \mathbb{E}_{s\sim\rho_i, \kappa_{i,t}\sim\phi_i(\cdot|s_{\mathcal{N}_{i,t}^{\kappa_{i,t-1}}})}[A_{\psi_i}^{\phi_i}(s_{\mathcal{N}_{i,t}^{\kappa_{i,t-1}}}, \kappa_{i,t})]$$

$$+ \mathbb{E}_{s\sim\rho_i, \kappa_{i,t}\sim\phi_i(\cdot|s_{\mathcal{N}_{i,t}^{\kappa_{i,t-1}}}), a_{i,t}\sim\pi_i(\cdot|s_{\mathcal{N}_{i,t}^{\kappa_{i,t-1}}},\kappa_{i,t})}[A_{\psi_i}^{\phi_i,\pi_i}(s_{\mathcal{N}_{i,t}^{\kappa_{i,t-1}}}, \kappa_{i,t}, a_{i,t}, a_{\mathcal{N}_{-i,t}^{\kappa_{i,t}}})]$$

$$- \mathbb{E}_{\kappa_{i,t}\sim\bar{\phi}_i(\cdot|s_{\mathcal{N}_{i,t}^{\kappa_{i,t-1}}})}[\eta'\rho^{\kappa_{i,t}+l}]$$

$$- \left[\nu_{\phi_i}D_{KL}^{max}(\psi_i, \bar{\psi}_i) + \nu_{\pi_i}D_{KL}^{max}(\psi_i, \bar{\psi}_i)\right]. \tag{56}$$

Let

$$L_{\psi_i}(\bar{\psi}_i) = \mathbb{E}_{s\sim\rho_i, \kappa_{i,t}\sim\phi_i(\cdot|s_{\mathcal{N}_{i,t}^{\kappa_{i,t-1}}})}\left[A_{\psi_i}^{\phi_i}(s_{\mathcal{N}_{i,t}^{\kappa_{i,t-1}}}, \kappa_{i,t})\right]$$

$$+ \mathbb{E}_{\substack{s\sim\rho_i \\ \kappa_{i,t}\sim\phi_i(\cdot|s_{\mathcal{N}_{i,t}^{\kappa_{i,t-1}}}) \\ a_{i,t}\sim\pi_i(\cdot|s_{\mathcal{N}_{i,t}^{\kappa_{i,t-1}}},\kappa_{i,t})}}\left[A_{\psi_i}^{\phi_i,\pi_i}(s_{\mathcal{N}_{i,t}^{\kappa_{i,t-1}}}, \kappa_{i,t}, a_{i,t}, a_{\mathcal{N}_{-i,t}^{\kappa_{i,t}}})\right]$$

$$M_{KL} = \nu_{\phi_i}D_{KL}^{max}(\phi_i, \bar{\phi}_i) + \nu_{\pi_i}D_{KL}^{max}(\pi_i, \bar{\pi}_i),$$

we arrive at Proposition 2.

## C.5 THE PROOF OF LEMMA 2

*Proof.* From Theorem 1 in TRPO, we have the following result for the policy $\psi_i$ of agent $i$:

$$J_c(\bar{\psi}_i) - J_c(\psi_i) \leq L_{\psi_i}(\bar{\psi}_i) - \frac{4\alpha^2\gamma \max_{s,\kappa_{i,t}} |A_{c,\psi_i}^{\bar{\phi}_i}(s_{\mathcal{N}_{i,t}^{\kappa_{i,t-1}}}, \kappa_{i,t})|}{(1-\gamma)^2}$$

$$\alpha = D_{TV}^{max}(\psi_i, \bar{\psi}_i) = \max_s D_{TV}(\psi_i, \bar{\psi}_i). \tag{57}$$

Subsequently, by applying Pinsker's inequality (Fedotov et al., 2003) $D_{TV}(p,q)^2 \leq D_{KL}(p,q)$, we can obtain:

$$J_c(\bar{\psi}_i) - J_c(\psi_i) \leq L_{\psi_i}(\bar{\psi}_i) - \frac{4\gamma \max_{s,\kappa_{i,t}} |A_{c,\psi_i}^{\bar{\phi}_i}(s_{\mathcal{N}_{i,t}^{\kappa_{i,t-1}}}, \kappa_{i,t})|}{(1-\gamma)^2}D_{KL}^{max}(\psi_i, \bar{\psi}_i),$$

$$D_{KL}^{max}(\psi_i, \bar{\psi}_i) = \max_s D_{KL}(\psi_i, \bar{\psi}_i). \tag{58}$$

Note that the surrogate objective $L_{\psi_i}(\bar{\psi}_i)$ consists of two components: one related to communication and the other to decision-making. Since the communication cost depends only on the communication policy $\phi_i$ and not on the decision policy $\pi_i$, its gradient with respect to $\pi_i$ is zero. Consequently, when optimizing $\pi_i$ independently, the communication cost contributes nothing to the policy update. Since the communication cost is independent of $\pi_i$, its contribution to the change in the objective is zero.

Similarly,

$$D_{KL}^{\max}(\psi_i, \bar{\psi}_i) = \max_s D_{KL}(\psi_i, \bar{\psi}_i) = \max_s D_{KL}(\phi_i, \bar{\phi}_i). \tag{59}$$

Let $\nu_{c,\phi_i} = \dfrac{4\gamma \max_{s,\kappa_{i,t}} \left| A_{c,\psi_i}^{\bar{\phi}_i}(s_{\mathcal{N}_{i,t}^{\kappa_{i,t-1}}, \kappa_{i,t}}) \right|}{(1-\gamma)^2}$, we can obtain:

$$J_c(\bar{\psi}_i) - J_c(\psi_i) \leq L_{c,\psi_i}(\bar{\phi}_i) - \nu_{c,\phi_i} D_{KL}^{\max}(\phi_i, \bar{\phi}_i),$$

$$\nu_{c,\phi_i} = \frac{4\gamma \max_{s,\kappa_{i,t}} |A_{c,\psi_i}^{\bar{\phi}_i}(s_{\mathcal{N}_{i,t}^{\kappa_{i,t-1}}}, \kappa_{i,t})|}{(1-\gamma)^2}. \tag{60}$$

We arrive at Lemma 2.

### C.6 THE PROOF OF THEOREM 1

*Proof.* Based on Proposition 2 and Lemma 2, we can derive that, to improve reward performance while satisfying communication cost constraints, each agent must maximize its surrogate return, ensure that communication cost remains below a given threshold, and restrict policy updates within a local neighborhood—measured in terms of the maximum KL divergence.

First, for the communication cost upper bound constraint on each agent, we have:

$$J_c(\psi_i) + L_{c,\psi_i}(\bar{\phi}_i) - \nu_{c,\phi_i} D_{KL}^{\max}(\phi_i, \bar{\phi}_i) \leq c_i, \tag{61}$$

we can obtain:

$$D_{KL}^{\max}(\phi_i, \bar{\phi}_i) \geq \frac{J_c(\psi_i) + L_{c,\psi_i}(\bar{\phi}_i) - c_i}{\nu_{c,\phi_i}}. \tag{62}$$

Let $\mu_1 = J_c(\psi_i) + L_{c,\psi_i}(\bar{\phi}_i) - c_i$. Substituting Equation (62) into $M_{\mathrm{KL}}$, we obtain that the KL constraint between the old and new policies should be defined as:

$$\delta = \frac{\nu_{c,\phi_i}\mu_1 + \nu_{\phi_i}\nu_{\pi_i} D_{KL}^{max}(\pi_i, \bar{\pi}_i)}{\nu_{\phi_i}}, \tag{63}$$

where $M_{KL} = \nu_{\phi_i} D_{KL}^{\max}(\phi_i, \bar{\phi}_i) + \nu_{\pi_i} D_{KL}^{\max}(\pi_i, \bar{\pi}_i)$,

$$\eta' = \frac{2\bar{r} + (1-\gamma)G}{1-\gamma}, \sigma = \gamma,$$

$$\nu_{c,\phi_i} = \frac{4\gamma \max_{s,\kappa_{i,t}} |A_{c,\psi_i}^{\bar{\phi}_i}(s_{\mathcal{N}_{i,t}^{\kappa_{i,t-1}}}, \kappa_{i,t})|}{(1-\gamma)^2},$$

$$\nu_{\phi_i} = \frac{4\gamma \max_{s,\kappa_{i,t}} |A_{\psi_i}^{\phi_i}(s_{\mathcal{N}_{i,t}^{\kappa_{i,t-1}}}, \kappa_{i,t})|}{(1-\gamma)^2},$$

$$\nu_{\pi_i} = \frac{4\gamma \max_{s,\kappa_{i,t}a_{i,t}} |\widetilde{A}_{\psi_i}^{\phi_i,\pi_i}(s_{\mathcal{N}_{i,t}^{\kappa_{i,t-1}}}, \kappa_{i,t}, a_{i,t}, a_{\mathcal{N}_{-i,t}^{\kappa_{i,t}}})|}{(1-\gamma)^2},$$

$$\mu_1 = J_c(\psi_i) + L_{c,\psi_i}(\bar{\phi}_i) - c_i,$$

$$\delta = \frac{\nu_{c,\phi_i}\mu_1 + \nu_{\phi_i}\nu_{\pi_i} D_{KL}^{max}(\pi_i, \bar{\pi}_i)}{\nu_{\phi_i}}.$$

We arrive at Theorem 1.

---

**Algorithm 1** ACR-PPO

---

**Input**: Step size $\alpha_\theta, \alpha_\lambda$, batchsize B, number of agents $n$, number of episodes $Z$, steps per episode $T$, discount factor $\gamma$.

**Initalize**: Agent's network $\theta_1, ..., \theta_n$ (including communication policy network $\theta_1^\phi, ..., \theta_n^\phi$ and behavior policy network $\theta_i^\pi, ..., \theta_n^\pi$), $\forall i \in \mathcal{N}$, replay buffer $\mathcal{B}$.

1: **for** $t = 0 : T - 1$ **do**
2:    **for** $i = 1 : n$ **do**
3:       Select communication range $\kappa_{i,t} \sim \phi_i(\cdot | s_{\mathcal{N}_{i,t}^{\kappa_{i,t-1}}})$.
4:       Select action $a_{i,t} \sim \pi_i(\cdot | s_{\mathcal{N}_{i,t}^{\kappa_{i,t-1}}}, \kappa_{i,t})$.
5:    **end for**
6:    Push $\{< s_{\mathcal{N}_{i,t}^{\kappa_{i,t-1}}}, \kappa_{i,t}, a_{i,t}, r_{i,t}, s_{\mathcal{N}_{i,t+1}^{\kappa_{i,t}}} >, \forall i \in \mathcal{N}, t \in T\}$ into $\mathcal{B}$.
7:    **for** $i = 1 : n$ **do**
8:       Initialize policy parameter $\theta_i^\phi, \theta_i^\pi$ and Lagrangian multipliers $\lambda_i, \forall \in \mathcal{N}$.
9:       Compute parameters $\eta', \sigma, \nu_{c,\phi_i}, \nu_{\phi_i}, \nu_{\pi_i}, \mu_1, \delta$.
10:      Combine the advantage function and the cost advantage function, and update policy according to Equation(67).
11:   **end for**
12: **end for**

---

## C.7 THE DETAIL OF ALGORITHM 1

We focus on the practical implementation of the policy update in Theorem 1. Let $\theta_i$ denote the parameters of policy $\psi_i$, with $\theta_i^\phi$ and $\theta_i^\pi$ corresponding to the communication policy $\phi_i$ and the behavior policy $\pi_i$, respectively. For notational convenience, let $d_i = J_c(\bar{\psi}_i) - c_i$. $\lambda_i$ is a scalar variable. We apply Lagrangian relaxation to the communication cost constraint, yielding:

$$\max_{\theta_i^\phi, \theta_i^\pi} \min_{\lambda_i > 0} \left[ \mathbb{E}_{s, \kappa_{i,t}, a_{i,t}} \left[ \widetilde{A}_{\theta_i}^{\theta_i^\phi, \theta_i^\pi} \left( s_{\mathcal{N}_{i,t}^{\kappa_{i,t-1}}}, \kappa_{i,t}, a_{i,t}, a_{\mathcal{N}_{-i,t}^{\kappa_{i,t}}} \right) \right] \right.$$
$$\left. + \mathbb{E}_{s, \kappa_{i,t}} A_{\theta_i}^{\theta_i^\phi} \left( s_{\mathcal{N}_{i,t}^{\kappa_{i,t-1}}}, \kappa_{i,t} \right) - \lambda_i \left( \mathbb{E}_{s, \kappa_{i,t}} \left[ A_{c,\theta_i}^{\theta_i^\phi} \left( s_{\mathcal{N}_{i,t}^{\kappa_{i,t-1}}}, \kappa_{i,t} \right) \right] + d_i \right) \right], \quad (64)$$
$$\text{s.t.} \quad M_{KL} \le \delta,$$

where $\lambda_i$ is a scalar variable. Let

$$A_{\theta_i}^{\lambda_i, \theta_i^\phi, \theta_i^\pi} = \widetilde{A}_{\theta_i}^{\theta_i^\phi, \theta_i^\pi} \left( s_{\mathcal{N}_{i,t}^{\kappa_{i,t-1}}}, \kappa_{i,t}, a_{i,t}, a_{\mathcal{N}_{-i,t}^{\kappa_{i,t}}} \right) + A_{\theta_i}^{\theta_i^\phi} (s_{\mathcal{N}_{i,t}^{\kappa_{i,t-1}}}, \kappa_{i,t})$$
$$- \lambda_i \left( A_{c,\theta_i}^{\theta_i^\phi} (s_{\mathcal{N}_{i,t}^{\kappa_{i,t-1}}}, \kappa_{i,t}) + d_i \right),$$

$$(65)$$

we can obtain:

$$\max_{\theta_i^\phi, \theta_i^\pi} \min_{\lambda_i > 0} \left[ \mathbb{E}_{s, \kappa_{i,t}, a_{i,t}} A_{\theta_i}^{\lambda_i, \theta_i^\phi, \theta_i^\pi} \right], \quad \text{s.t } M_{KL} \le \delta. \quad (66)$$

After rearranging the terms, we can obtain:

$$\max_{\theta_i^\phi, \theta_i^\pi} \min_{\lambda_i > 0} \mathbb{E}_{s, \kappa_{i,t}, a_{i,t}} \left[ \min \left( U_i, (1 \pm \epsilon) U_i \right) A_{\theta_i}^{\lambda_i, \theta_i^\phi, \theta_i^\pi} \right], \quad (67)$$

where $U_i = \frac{\bar{\phi}_i(\theta_i^\phi) \cdot \bar{\pi}_i(\theta_i^\pi)}{\phi_i(\theta_i^\phi) \cdot \pi_i(\theta_i^\pi)}$ is the importance sampling ratio, measuring the updated joint policy against the current policy. This ratio is clipped within the interval $[1 - \epsilon, 1 + \epsilon]$, consistent with the standard PPO clipping mechanism. Each agent $i$ updates $\theta_i^\phi$ and $\theta_i^\pi$ by maximizing the surrogate objective in Equation (67), allowing the agent to simultaneously maximize the expected return while minimizing its communication range.

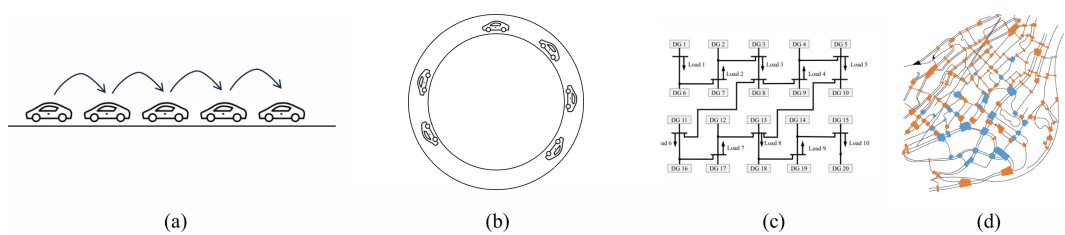

(a)         (b)         (c)         (d)

Figure 3: The environments. (a) CACC (Catchup), where the objective is to adaptively coordinate a fleet of ehicles to minimize spacing and speed disturbances based on real-time V2V communication. (b) Flow (Ring), which shares similar objectives with CACC—maintaining target speed and avoiding collisions. (c) IEEE PowerGrid (PowerGrid), where the objective is to improve voltage control performance in an islanded microgrid system. (d) ATSC (Monaco), where the objective is to dynamically adjust traffic signal phases based on real-time traffic measurements to reduce congestion.

# D  EXPERIMENTS

## D.1  EXPERIMENTS

The simulation environment setup is shown in Figure 3.

**CACC** (Chu et al., 2020). For both CACC tasks, we simulate a platoon of 8 vehicles over a 60-second period with a 0.1-second control interval. Each vehicle observes and shares its headway ($h$), velocity ($v$), and acceleration ($a$) with neighbors within a two-step communication range. The safety constraints are defined as: $h \geq 1$m, $v \leq 30$m/s, and $|a| \leq 25$m/s$^2$. While safe RL is relevant in this context, it represents a substantial topic that falls outside the scope of this paper. Therefore, we employ a simple heuristic optimal velocity model (OVM) to perform longitudinal vehicle control under the aforementioned constraints. The OVM behavior is influenced by hyper-parameters: headway gain ($\alpha$), relative velocity gain ($\beta$), stop headway ($h_{st} = 5$m), and full-speed headway ($h_{go} = 35$m). Typically, $\alpha$ and $\beta$ represent human driver behavior; however, in this work, we train NMARL to recommend appropriate ($\alpha, \beta$) pairs for each OVM controller. These recommendations are selected from four predefined levels: $(0, 0)$, $(0.5, 0)$, $(0, 0.5)$, and $(0.5, 0.5)$. Assuming the target headway and velocity profile are $\bar{h} = 20$m and $v_t$, respectively, the cost function for each agent is defined as $(h_{it} - \bar{h})^2 + (v_{it} - v_t)^2 + 0.1u_{it}^2$. When a collision occurs ($h_{it} < 1$m), a substantial penalty of 1000 is imposed on each agent, and the state becomes absorbing. Additionally, a collision avoidance cost of $5(2h_{st} - h_{it})^2$ is incorporated during training to penalize potential collision scenarios.

**Flow** (Vinitsky et al., 2018). (a) The figure eight network, serves as a closed-loop representation of an intersection. Within this figure eight network comprising 14 vehicles in total, we observe queue formation due to simultaneous vehicle arrivals at the intersection, causing vehicles to decelerate in compliance with right-of-way regulations. This phenomenon substantially decreases the average vehicle speed throughout the network. In a mixed-autonomy environment, a subset of vehicles functions as CAVs tasked with managing traffic flow through the intersection to enhance system-wide velocities. The MDP components for this benchmark are specified as follows. (b) The grid network serves as an idealized model of grid-like urban environments, such as Manhattan. There are 25 agents in this environment. This problem formulation is designed to emphasize challenges in traffic light control coordination, specifically addressing issues of partial observability and the scalability of RL algorithms with respect to action dimensionality. Resolving this problem will lead to the development of novel traffic light control strategies that reduce average per-vehicle delay while maintaining a degree of fairness across the network.

**IEEE PowerGrid** (Chen et al., 2021). The IEEE PowerGrid simulation platform is constructed based on the line and load specifications and there are 40 agents in this environment. To better represent real-world power systems, random load variations are introduced across the entire microgrid with 20% perturbations from the nominal values specified. At each simulation step, random disturbances within 5% of the nominal values for each load are also added to simulate the stochastic

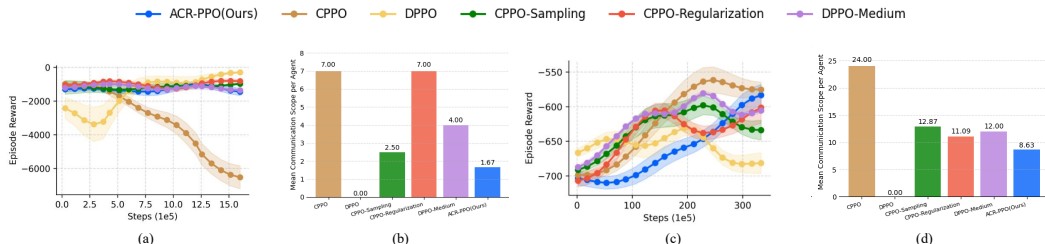

Figure 4: The policy performance and communication range per agent. (a) The policy performance in Slowdown. (b) The average communication range per agent in Slowdown. (c) The policy performance in Grid. (d) The average communication range per agent in Grid. Experimental results show that our method can maintain performance as much as possible while adaptively reducing the communication range, thereby lowering costs.

disruptions present in real-world power grids. The DGs are controlled with a sampling time of 0.05s, and each DG can communicate with its neighbors via local communication edges.

**ATSC** (Chu et al., 2020). For both scenarios, each episode simulates peak-hour traffic, and a 5s control interval is applied to prevent traffic lights from switching too frequently, based on RL control latency and driver response delay. Thus, one MDP step corresponds to 5s simulation and the horizon is 720 steps. Further, a 2s yellow time is inserted before switching to red light for safety purposes. In ATSC, the real-time traffic flow, that is, the total number of approaching vehicles along each incoming lane, is measured by near-intersection induction-loop detectors (ILDs). The cost of each agent is the sum of queue lengths along all incoming lanes. The Ring scenario has 22 agents, while the Monaco scenario has 28 agents. To validate the performance of our method in larger-scale scenarios, we additionally adopt an ATSC-NewYork environment, which simulates traffic signal control in New York City and involves 436 agents.

**Communication Cost**. For consistency in evaluation and ease of computation, all experiments adopt this setting. See Appendix B.3.

### D.2 BASELINE

In addition to CPPO and DPPO, we introduce two additional baseline methods for comparison: CPPO-Sampling and CPPO-Regularization.

**CPPO-Sampling**. This method randomly masks a subset of the observation inputs to the agent's critic, simulating random sampling from a fixed distribution. Notably, in ACR-PPO, the communication cost (see Equation 42) is strongly correlated with the parameter $\kappa$. To ensure consistency in our evaluation metric, we adopt a simplified model of multi-hop communication: we assume that messages propagate outward in wave-like fashion within the communication range, and any agent whose observation is not masked is considered reachable and thus allowed to communicate. Under this assumption, we approximate the average ID of masked agents as the effective communication hop count. To emulate a stringent communication constraint, we set the masking probability to 80%, meaning each agent receives observations from others with only a 20% probability.

**CPPO-Regularization**. L1 Regularization is another common approach to enforcing communication constraints, aiming to reduce communication overhead while preserving policy performance. To this end, we introduce an additional regularization term on the critic's input to explicitly discourage unnecessary communication. The specific formulation is as follows:

$$Loss_{tot} = Loss_{CPPO} + \lambda L_1(Input). \tag{68}$$

### D.3 THE IMPLEMENT DETAIL OF OUR METHODS

**Framework**. The overall architecture of the algorithm is illustrated in the figure. **Each agent** maintains two sub-policies that are updated sequentially: the communication policy first dynamically

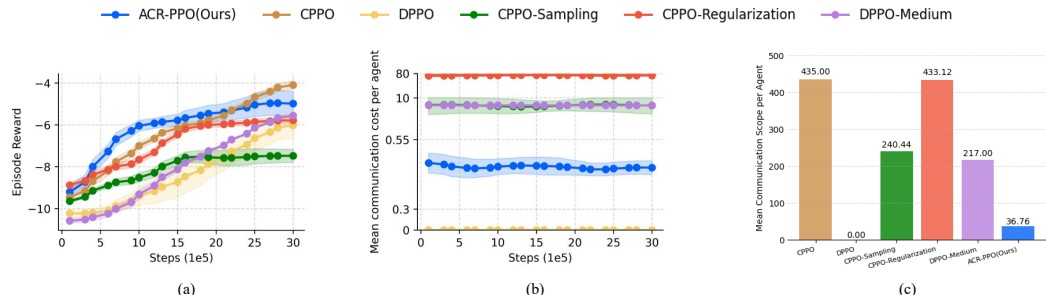

Figure 5: Experiments conducted in a larger-scale traffic scenario, "ATSC-NewYork (Agent number=436)". (a) Performance of the experiment. (b) Average communication cost per agent. (c) Average communication range per agent. The results show that our method remains applicable to larger-scale problems and effectively reduces communication costs.

determines the agent's communication range, and the behavior policy then selects actions based on neighboring agents within that range.

**Model Size.** Due to varying observation dimensions across environments, we take the Catchup environment as an example(agent number $= 16$, obs_dim $= 30$, action_dim $= 5$):

- Policy net per agent: $30 \times 64 + 64 \times 64 + 64 \times 5 \approx 6{,}000$ params
- Value net per agent : $30 \times 64 + 64 \times 64 + 64 \times 1 \approx 6{,}000$ params
- Communication Policy net per agent: $30 \times 64 + 64 \times 64 + 64 \times 16 \approx 7{,}000$ param
- Communication Value net per agent : $30 \times 64 + 64 \times 64 + 64 \times 1 \approx 6{,}000$ params
- **Total:** $\approx$**250,000** params per agent.

**HyperParameters Setting**. Table 1 presents the parameters associated with ACR. Additional pertinent parameters follow the conventions of CACC, Flow, ATSC and IEEE PowerGrid.

**Training Time**. Table 2 shows the training time of the overall algorithm. We observe that training time increases primarily because our method employs a parameter-sharing mechanism, which requires a uniform batch size and leads to longer training time.

**Computing Setting**. We conducted our experiments on a system equipped with an Intel(R) Core i9-13900K processor (8 performance cores and 16 efficiency cores, totaling 32 threads) with a base frequency of 3.0 GHz. The system is integrated with one NVIDIA GeForce RTX 4090 GPU to accelerate the training process. The operating system is Ubuntu 20.04 LTS.

**Computational Complexity**. Based on the description in Algorithm 1, the time complexity of ACR is $\mathcal{O}(T \cdot N \cdot H \cdot (M + P))$, where: $T$ denotes the total number of timesteps, $N$ is the number of agents, $H$ represents the number of PPO epochs per update, $M$ is the number of parameters in the communication policy, $P$ is the number of parameters in the behavior policy. In terms of communication cost, our method achieves adaptive control by dynamically restricting the communication range through the communication policy. Specific experimental results are shown in the second row of Figure 1.

## D.4 MORE RESULTS

More experimental results are shown in Figure 4. We have added additional experimental results for the Slowdown and Grid environments. The results indicate: (1) Our method can minimize costs as much as possible by adaptively adjusting the communication range while maintaining performance. (2) The performance of CPPO under the Slowdown environment is not as good as that of DPPO. We speculate that this may be due to excessive communication leading to the generation of redundant information. (3) In Figure 4(a), our method closely approaches the performance of DPPO, whereas in Figure 4(c), our method closely approaches the performance of CPPO. This demonstrates that our method can effectively adapt to different environments.

### D.5 SCALABILITY EVALUATION

To further demonstrate the scalability of our approach, we evaluate on a "New York" environment comprising 436 agents, which simulates traffic light control in a real-world urban setting. The results are presented in the Figure 5. The results show that our algorithm scales effectively to larger scenarios, achieving lower communication costs while maintaining near-optimal performance.

### D.6 FURTHER ANALYSIS ON COMMUNICATION RANGE VARIATION

To further investigate how the communication range evolves during training, we conduct two additional experiments. First, we examine the communication range selection behavior of all agents across different episodes. As shown in Figure 6, we present results from Episode No.100, No.400, and No.800 in the PowerGrid environment (20-agent version). The results indicate that, during early training (Episode No.100), the algorithm is still exploring, and the average communication range is relatively high (approximately $\kappa = 9.42$), resembling random sampling behavior. As training progresses (Episode No.400), agents gradually reduce their communication range ($\kappa = 9.25$), and by the end of training (Episode No.800), the average range further decreases to $\kappa = 8.82$. Although this average reduction appears small, it is important to note that communication cost typically grows exponentially with the communication range. Therefore, even minor reductions in $\kappa$ can lead to significant savings in overall communication cost.

Second, we perform a more detailed analysis of individual agent communication patterns at the end of training. As shown in Table 3, we present the per-step communication range for each agent during Episode No.800. The results reveal that while the overall average communication range is $\kappa = 8.82$, individual agents exhibit considerable variation. Some agents (e.g., Agent 9, Agent 17, and Agent 2) maintain a communication range larger than the average, whereas others (e.g., Agent 12 and Agent 19) operate with significantly smaller ranges. We hypothesize that agents requiring larger communication ranges are often located at critical junctions or high-load nodes in the power grid, where coordination with neighboring agents is essential. In contrast, agents with smaller communication ranges are likely situated in peripheral areas, where local coordination suffices. This spatial variation in communication needs reflects the heterogeneous structure of the environment and highlights the adaptability of our algorithm in tailoring communication strategies to local requirements.

## E RELATED WORKS

### E.1 SEQUENTIAL UPDATING

Sequential updating is a significant approach for MARL policy iteration. HATRPO (Kuba et al., 2022), MAT (Wen et al., 2022), HARL (Zhong et al., 2024) and ACE (Li et al., 2023a) adopt a sequential update approach to iteratively update each agent's policy. However, these methods primarily focus on sequential updates at the agent level rather than on the policy iteration of individual. Other approaches, such as STEER (Zhang et al., 2024a), LtoS (Yi et al., 2022), and Bi-AC (Zhang et al., 2020), adopt a hierarchical structure to coordinate the policy updates of individual agents. However, these methods primarily focus on asynchronous action coordination, reward sharing, and Nash equilibrium computation. A method similar to ours is (Geng et al., 2024), which considers relay nodes as leaders and sources as followers. In contrast, our work does not focus on the grouping of relay nodes, but instead emphasizes the sequential update of policy execution within each agent's multi-hop communication range.

## F LIMITATION

Our work may have several limitations as follows. First, ACR requires more time for the search process. Compared to fixed $\kappa$-hop or graph structure-based sampling methods, our approach typically

Table 1: Hyper-parameters Setting of ACR.

| Parameter | Ring | Catchup | Monaco | PowerGrid |
|---|---|---|---|---|
| lambda | 0.5 | 0.5 | 0.5 | 0.5 |
| clip | 0.15 | 0.2 | 0.2 | 0.2 |
| target KL | 7.5e-3 | 0.01 | 7.5e-3 | 7.5e-3 |
| batch size | 256 | 256 | 256 | 256 |
| buffer size | 15 | 15 | 15 | 15 |
| gamma | 0.99 | 0.99 | 0.99 | 0.99 |
| learning rate | 5e-4 | 5e-5 | 5e-4 | 5e-5 |
| learning rate V | 5e-4 | 7e-4 | 5e-4 | 5e-4 |
| activation | ReLU | ReLU | ReLU | ReLU |
| nn hidden size | 64 | 64 | 64 | 64 |
| cost bound | 3000 | 1200 | 3000 | 3000 |

Table 2: Training Time in our experiments.(min)

| Algorithm | CPPO | DPPO-Medium | DPPO | PPO-Reg | PPO-Ramdom | ACR-PPO |
|---|---|---|---|---|---|---|
| Slowdown | 22±1 | 22±3 | 31±0 | 41±1 | 15±2 | 39±2 |
| Grid | 297±2 | 294±1 | 412±3 | 370±0 | 301±0 | 307±5 |
| Ring | 277±1 | 284±3 | 266±2 | 301±6 | 277±2 | 299±10 |
| Catchup | 20±5 | 25±2 | 21±1 | 41±1 | 20±0 | 25±4 |
| Monaco | 155±5 | 149±5 | 144±12 | 147±11 | 147±4 | 151±0 |
| PowerGrid | 94±0 | 109±0 | 94±1 | 105±1 | 108±1 | 105±2 |
| Eight | 105±2 | 92±5 | 78±1 | 120±16 | 104±1 | 106±3 |
| New York | 431±32 | 412±7 | 388±12 | 392±8 | 374±10 | 525±2 |

Table 3: The communication range of each agent under in Powergird (No.800 Episode, 20 agents version. The x-axis represents 20 time steps, and the y-axis represents agents. Agents are sorted according to their average communication range within the episode.

| Agent | 1 | 3 | 5 | 7 | 9 | 11 | 13 | 15 | 17 | 19 | 20 | Avg $\kappa$ |
|---|---|---|---|---|---|---|---|---|---|---|---|---|
| Agent 9 | 13.5 | 9.5 | 15 | 16 | 15.5 | 11.5 | 11.5 | 14.5 | 10.5 | 14 | 2.5 | 12.19 |
| Agent 17 | 12 | 9 | 12.5 | 12.5 | 8.5 | 10 | 16.5 | 13 | 12.5 | 4.5 | 11.5 | 11.14 |
| Agent 2 | 17.5 | 10.5 | 10.5 | 9 | 10 | 9.5 | 4 | 10.5 | 12 | 7 | 11.5 | 10.18 |
| Agent 10 | 11.5 | 6 | 9 | 6.5 | 12 | 11 | 14 | 16 | 4 | 11 | 5 | 9.64 |
| Agent 14 | 17.5 | 8 | 12 | 8 | 8.5 | 16 | 5.5 | 7 | 3 | 13.5 | 6 | 9.55 |
| Agent 11 | 8.5 | 8.5 | 10 | 11 | 10.5 | 7.5 | 12.5 | 5 | 5 | 10.5 | 14.5 | 9.41 |
| Agent 15 | 6.5 | 9.5 | 13 | 9.5 | 7 | 10 | 6.5 | 14 | 7 | 8.5 | 11.5 | 9.36 |
| Agent 1 | 17.5 | 8 | 6.5 | 9.5 | 8.5 | 4.5 | 3.5 | 14 | 11.5 | 9.5 | 9 | 9.27 |
| Agent 4 | 1 | 8 | 2 | 16.5 | 10 | 15.5 | 17.5 | 4 | 7.5 | 9.5 | 5.5 | 8.82 |
| Agent 18 | 9.5 | 0 | 12 | 6 | 11 | 5.5 | 9.5 | 4.5 | 9.5 | 12.5 | 15.5 | 8.68 |
| Agent 5 | 8 | 7 | 8 | 8.5 | 18 | 14 | 11 | 1.5 | 8.5 | 6 | 3 | 8.50 |
| Agent 16 | 8 | 5 | 6 | 6 | 9 | 12.5 | 11 | 16 | 5 | 4.5 | 8.5 | 8.31 |
| Agent 7 | 7 | 1 | 10.5 | 12 | 15.5 | 7.5 | 8 | 6 | 7.5 | 6 | 10 | 8.27 |
| Agent 20 | 3.5 | 10.5 | 7 | 12.5 | 1.5 | 1.5 | 7.5 | 11 | 12 | 12 | 11.5 | 8.23 |
| Agent 8 | 8 | 12 | 7.5 | 3.5 | 7 | 8.5 | 7.5 | 15 | 5.5 | 4.5 | 8 | 7.91 |
| Agent 3 | 8.5 | 4.5 | 4 | 9 | 10.5 | 14.5 | 12 | 4 | 7 | 3.5 | 8 | 7.77 |
| Agent 13 | 1.5 | 4 | 15 | 11.5 | 9 | 2 | 8 | 8 | 10.5 | 10.5 | 2.5 | 7.50 |
| Agent 6 | 2 | 4.5 | 15 | 1.5 | 5.5 | 8 | 8.5 | 9 | 6 | 11 | 10.5 | 7.41 |
| Agent 12 | 14 | 10 | 6 | 6.5 | 6.5 | 1.5 | 3 | 11.5 | 14.5 | 3 | 2 | 7.14 |
| Agent 19 | 9.5 | 1.5 | 10.5 | 13 | 2.5 | 8 | 5.5 | 8.5 | 2.5 | 12 | 5 | 7.14 |

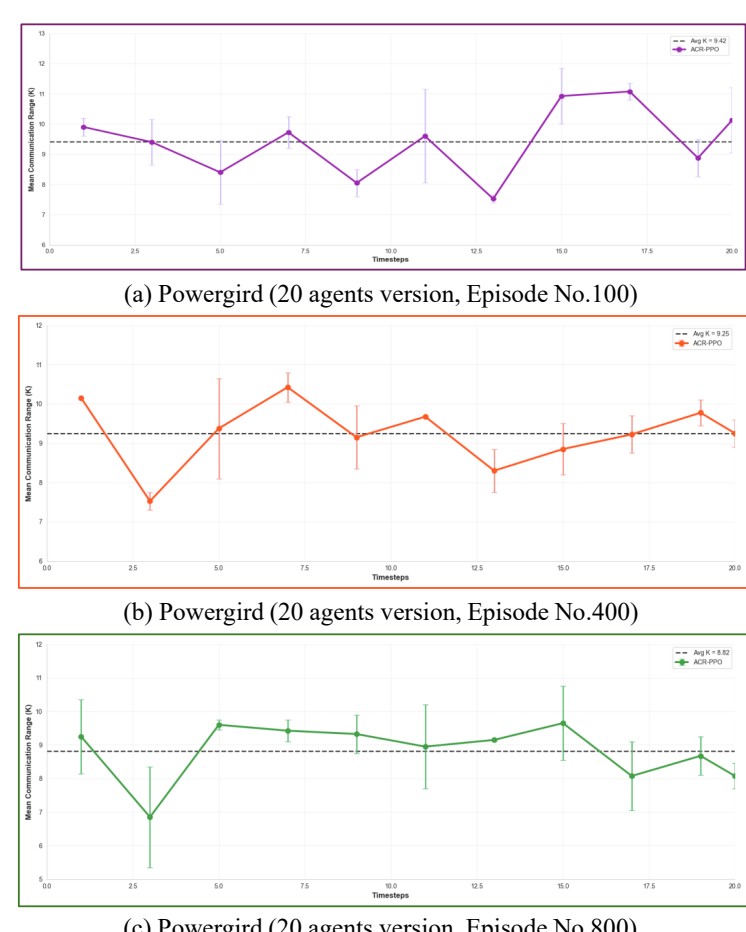

(a) Powergird (20 agents version, Episode No.100)

(b) Powergird (20 agents version, Episode No.400)

(c) Powergird (20 agents version, Episode No.800)

Figure 6: Performance of the average communication range of agents across episodes. We extract the training performance from three fixed episodes throughout the entire training process: (a) early training (Episode No.100), (b) mid-training (Episode No.400), and (c) late training (Episode No.800). The results show that, over time, the average communication range gradually decreases due to the constraints imposed by our method's policy. Although these reductions are small, considering that this represents the average across all agents and given the exponential nature of communication costs, even minor reductions in communication range lead to significant decreases in overall communication cost.

demands additional computational resources and longer running times to find a suitable communication range in different timestep. Second, ACR relies on the assumption that the distribution of communication policies before and after updates remains approximately unchanged. This limits the effectiveness of our method in scenarios where the communication range changes abruptly or frequently. Moreover, this paper focuses on adaptive communication range selection and cost optimization among agents, and does not model the potential internal delays introduced by sequential update mechanisms. Although such delays have not been systematically studied in the MARL field and are not the core focus of our work, we have added a discussion of the potential impacts of such temporal dependencies in Appendix D.3, aiming to provide insights for future work.

## G  THE USE OF LARGE LANGUAGE MODELS

Large language models were used only for the language expression polishing and grammar correction of this paper, in order to ensure the writing accurate, fluent, and conform to academic norms.

