# OpenReview forum: "Scalable Cooperative Multi-Agent Reinforcement Learning with Adaptive Communication Range"
_ICLR.cc/2026/Conference — Submitted to ICLR 2026_

### Official Review · Reviewer_4vHw · 2025-10-28

**Soundness:** 2
**Presentation:** 3
**Contribution:** 2
**Rating:** 6
**Confidence:** 3

**Summary:**

This paper introduces **ACR-PPO**, a networked multi-agent reinforcement learning (MARL) method that incorporates communication among agents. The authors provide a theoretical analysis of how the communication range affects performance improvement and evaluate the proposed method on several cooperative tasks. The results suggest that ACR-PPO can significantly reduce communication cost while achieving comparable performance in some tasks.

**Strengths:**

1. The paper is well-written and clearly structured.
2. It provides a theoretical analysis of the communication range under communication cost, which could be of interest to researchers in the MARL with communication community.

**Weaknesses:**

Overall, the novelty of the paper is not clear to me, particularly regarding the discussion of the “global state.” While the paper attempts to connect its approach to Networked MARL with communication, the discussion of related work is limited. The differences between this paper and the latest achievements in the Networked MARL literature are not clear.

Moreover, the GACML and AC2C methods mentioned by the authors have already employed local information for determining communication (although they employ centralized critics only during training), which contradicts the authors' claim "_these methods typically require the global state_".

The baseline comparison does not include communication-based methods such as GACML and AC2C. Without these comparisons, it is difficult to assess the effectiveness of the proposed method.

In Figure 1, the definition of communication cost is unclear. PPO itself does not involve communication among agents, so its communication cost should be zero. Why would communication be necessary if a regularized PPO variant (PPO + regularization) can outperform ACR-PPO on the presented tasks?


**Other Comments**

- **Line 43:** Mean-field MARL does not typically involve or claim explicit communication among agents. Why is it included in this introduction and related work?
- **Line 101:** Please clarify which appendix is being referenced.
- **Definition 1:** This definition is said to follow Qu et al., but the assumptions and meaning underlying it are not clearly stated.
- **Equations (10) and (11):** These are somewhat confusing.
    - In Eq. (10), since ki,t is given, it is unclear why it appears in the sampling process.
    - In Eq. (11), if Q is replaced using Eq. (10), both s and ki,t​ appear to be sampled twice.

**Questions:**

See weakness

---

> ### Author Response · Authors · 2025-11-25
> **Reply1**
>
> We sincerely apologize for any confusion caused and would like to provide further clarification on this issue.
>
> ## W1 / W2 / W3: the novelty of the paper ... not clear.
>
> We sincerely apologize for the confusion. We would like to reframe our contribution from the perspective of "Global state." Current MARL training methods have fundamental limitations in networked, large-scale systems:
> - Centralized Training Methods (e.g., CTDE/CTCE): These require the **global state-action value function** $Q(\mathbf{s}, \mathbf{a})$, incurring high communication costs that are impractical for large-scale systems. For example, Gated-ACML [1] and AC2C [2] still rely on the global state-action value function $Q(\mathbf{s}, \mathbf{a})$ during the pre-training phase to generate supervision signals indicating whether communication is beneficial or not.
> - Fully Decentralized Training Methods (DTDE): These rely solely on the **local state-action value function** $Q_i(s_i, a_i)$ such as DPPO. avoiding communication but suffering from non-stationarity and degraded performance.
> - Our work addresses the Networked MARL setting where agents communicate only with neighbors within a $\kappa$-hop range[3] [4] [5] . This method lies between centralized and decentralized training methods. Under the spatial exponential decay assumption, we leverage a **truncated value function** $Q(\mathbf{s}^{\kappa}, \mathbf{a}^{\kappa})$ **that only depends on local and $\kappa$-hop neighbor information**. This allows for scalability while retaining necessary cooperation information. Although spatial decay validates the potential for effective cooperation with local information, determining the optimal communication range is a critical open problem. Increasing the range improves performance but leads to an exponential increase in communication cost.
>
> **Our Novelty:** (1) We propose a scalable MARL method that formulates communication decisions as a constrained optimization problem under a budget, enabling the adaptive adjustment of the communication range. (2) We **define a time-varying truncated value function and derive optimization objectives** by alternately updating the communication and behavior policies for each agent. (3) We provide a **theoretical guarantee** for achieving monotonic performance improvement while strictly satisfying communication budget constraints.
>
> ## W4: In Figure 1 ... tasks?
>
> ① Regarding the definition of communication cost, we have placed it in **Appendix B.3**.
> ② We sincerely apologize for any inconvenience caused. By "regularization variants" we refer to those based on CPPO. The PPO you mentioned as "not involving communication" corresponds to our DPPO. We have updated the illustration in **Fig. 1** and the **Appendix D.2** baseline accordingly.
> ## Other Comments 1: Line 43: Mean-field MARL ... related work?
>
> Thank you very much for your valuable feedback. We have carefully revised the inaccuracies in the Introduction and Related Work sections in the latest version of the manuscript.
> ## Other Comments 2: Line 101: Please clarify ... referenced.
>
> Thank you for pointing this out. We have corrected the citation error in Line 101 and updated the reference to point to **Appendix B.3**.
> ## Other Comments 3: Definition 1.
>
> Thank you for your valuable suggestion. We have revised and clarified Definition 1 in the latest version, with detailed additions provided in the **Appendix B1 and B2**.
> ## Other Comments 4: Equations (10) and (11)
>
> We sincerely apologize for the confusion. To improve clarity, we have revised Definition 2 in new vision.
>
> ## Reference
>
> [1] Mao H, Zhang Z, Xiao Z, et al. Learning agent communication under limited bandwidth by message pruning. In AAAI 2020.
>
> [2] Wang X, Li X, Shao J, et al. AC2C: Adaptively controlled Two-Hop Communication for Multi-Agent Reinforcement Learning. In AAMAS 2023.
>
> [3] Qu G, Wierman A, Li N. Scalable reinforcement learning for multiagent networked systems. Operations Research, 2022, 70(6): 3601-3628.
>
> [4] Ma C, Li A, Du Y, et al. Efficient and scalable reinforcement learning for large-scale network control. Nature Machine Intelligence, 2024, 6(9): 1006-1020.
>
> [5] Zhang L, Li L, Wei W, et al. Scalable constrained policy optimization for safe multi-agent reinforcement learning. In NeurIPS 2024.

---

### Official Review · Reviewer_jnKk · 2025-10-31

**Soundness:** 1
**Presentation:** 3
**Contribution:** 3
**Rating:** 2
**Confidence:** 5

**Summary:**

This paper proposes Adaptive Communication Range PPO (ACR-PPO), a scalable cooperative multi-agent reinforcement learning algorithm that adaptively adjusts each agent’s communication range under a cost budget. It is based on the spatial correlation decay concept proposed in the literature, and different from Qu 2020, it uses a PPO formulation as the objective. It formulates communication-aware decision-making as a constrained optimization problem and jointly optimizes a communication policy and a behavior policy. The paper provides theoretical guarantees of monotonic performance improvement even with limited communication. Experiments on traffic, power-grid, and vehicle-platooning benchmarks show that ACR-PPO maintains near-centralized performance while significantly reducing communication cost.

**Strengths:**

-	The definition of the time-varying range Q/V function is novel
-	The decomposition of the policy into a range-selection and then a control part is also novel.
-	The proposed approach has diverse and interesting applications which are of value to this community.

**Weaknesses:**

My major concern is Lemma 1. Firstly, Lemma 1 does not say what rho is (which I believe should be gamma, the discounting factor, by looking at the proof). Then, looking at the proof in eq. (31), the second equality uses $\rho_{i,t} = \rho_{i,t}’$ for $t\leq \kappa$. This is not something that can be “required”. I think this is only true, if in the transition (eq. 1) and policy factorization, each agent only depends on 1-hop neighbors. If, for example, the transition and policy factorization depends on 2-hop neighbors, then $\rho_{i,t} = \rho_{i,t}’$ will be true up to $t\leq \kappa/2$ as each step the difference will propagate for 2 hops. Now that the authors are using a $\kappa$-hop dependence in the transition and policy, then $\rho_{i,t} = \rho_{i,t}’$ is only true up to t=1.

I believe this is a major flaw of the paper. While I found the concepts to be interesting, I would suggest fixing this flaw before the next submission.

**Questions:**

See weakness.

---

> ### Author Response · Authors · 2025-11-25
> **Reply1**
>
> We sincerely apologize for any confusion caused and would like to provide further clarification on this issue.
>
> We realize the confusion stemmed primarily from our failure to provide a clear and detailed explanation of the properties of spatial exponential decay, even though we cited Qu et al. [1] in Section 2.2. The explanation of the state dynamics Equation (1) was also omitted. This equation is simply a definition of the rate at which influence propagates and could adopt an arbitrary $\kappa$-hop setting. For theoretical convenience, previous works [1] [2] typically defaulted to $\kappa = 1$. We emphasize that the formulation and derivation remain valid regardless of this choice. A higher "propagation rate per unit time" merely accelerates convergence toward the $\kappa$-hop neighborhood, which would require a corresponding adjustment in the theoretical proof to maintain consistency.
>
> **About the spatial exponential decay.** The Exponential Decay Property, also known as Spatial Correlation Decay, states that the mutual impact of agents diminishes exponentially with their graph distance. This well-established property is widely used across multi-agent systems to design scalable, distributed algorithms (e.g., in power networks [3], epidemiological dynamics [4], and traffic flow networks [5]). To our knowledge, Qu et al. [1] were the first to introduce this concept to MARL. They formalized it into two assumptions, leading to the development of methods like [2] and [6]. Since then, subsequent studies, including those on decentralized safe MARL [7], have adopted this powerful framework.
>
> **We have re-stated these two assumptions in [8]:**
>
> **Assumption 1.** (Spatial Decay of Correlation for the Dynamics) Assume that there exist $\beta > 0$ , for any agents $i, j \in \mathcal{N}$, such that:
> $$
> max_{i\in \mathcal{N}}\sum_{j\in \mathcal{N}} e^{\beta d(i, j)}W^{ij} \leq \zeta,
> $$
> where $W^{ij}=\sup\|P^i(\cdot|z^j,z^{-j})-P^i(\cdot|z^j,z^{-j})\|_1$, $z_j=(s_j,a_j)$ and $z_j'=(s_j', a_j')$ represent two different state-action pairs of the agent $j$ respectively,  $z_j$ represents the state-action pair of the agent other than $j$. The value of $W^{ij}$ reflects the extent to which the local transition probability of agent $i$ is affected by the state and action of agent $j$. $d(i,j)$ represents the distance between agent $i$ and agent $j$，and $\delta\in[0,2/\gamma)$ is a constant. The formula shows that as $d(i,j)$ increases, $e^{\beta d(i,j)}$ grows exponentially since $\delta$ is a constant, rapidly forcing $W^{i,j} \to 0$.
>
> **Assumption 2.** (Spatial Decay of Correlation for the Policies) Assume that there exist $\xi, \beta \geq 0$ such that for any agent $i \in \mathcal{N}$, $\mathbf{s}_{\mathcal{N}\_{\kappa}^i} \in \mathcal{S}\_{\mathcal{N}\_{\kappa}^i}$, $\mathbf{s}\_{\mathcal{N}\_{\kappa}^{-i}}$, $\mathbf{s}\_{\mathcal{N}\_{\kappa}^{-i}}^{\prime} \in \mathcal{S}\_{\mathcal{N}\_{\kappa}^{-i}}$, one have:
>
> $$
> \sup_{\mathbf{s}_{\mathcal{N}\_\kappa^i}, \mathbf{s}\_{\mathcal{N}\_\kappa^{-i}}, \mathbf{s}\_{\mathcal{N}\_\kappa^{-i}}^{\prime}} \left|\pi^i\left(\cdot|\mathbf{s}\_{\mathcal{N}\_\kappa^i}, \mathbf{s}\_{\mathcal{N}\_\kappa^{-i}}\right) -\pi^i\left(\cdot|\mathbf{s}\_{\mathcal{N}\_\kappa^i}, \mathbf{s}\_{\mathcal{N}\_\kappa^{-i}}^{\prime}\right)\right| \leq \xi e^{-\beta \kappa}.
> $$
>
> Based on the two assumptions above, we can naturally derive the exponentially decaying form of $Q$. Due to OpenReview’s character limit, we have added the full derivation details in **Appendix B1.**

---

> ### Author Response · Authors · 2025-11-25
> **Reply2**
>
> ## Overall, we have made changes in the following parts in new Submission:
> - First, **in Section 2 Preliminaries, we added a footnote**:  “Typically, this $\kappa$ can be any fixed value. This primarily reflects the agents' dependence on state transitions. However, to facilitate subsequent derivations, this paper sets its propagation rate to 1 by default.” This helps prevent potential misunderstandings about the state transition dynamics.
> - Second, **in Appendix B1**, we added a dedicated paragraph at the beginning to re-explain the key properties of spatial exponential decay, which provides deeper insight into our method. Furthermore, we have updated the presentation in Section B.1 Proof of Lemma 1. Specifically, following the statement: "We require $\rho_{i,t} = \rho_{i,t}'$ for all $t \leq \kappa$,”  we inserted the clarification:  "The reason is that, due to the local dependence structure in Equation (1) and the spatial decay assumptions (Assumption 1 and Assumption 2), if the range of neighborhood dependence changes in Equation (1), 'the propagation rate per unit time' must be adjusted accordingly. For convenience in analysis, it is typically set to 1.”
>
> We sincerely thank you for your valuable feedback and hope these clarifications address your concerns. We kindly ask you to reconsider your evaluation in light of these revisions.
>
>
> ## Reference
>
> [1] Qu G, Wierman A, Li N. Scalable reinforcement learning for multiagent networked systems. Operations Research, 2022, 70(6): 3601-3628.
>
> [2] Zhang Y, Qu G, Xu Pan, et al. Global Convergence of Localized Policy Iteration in Networked Multi-Agent Reinforcement Learning. In ACM SIGMETRICS 2023.
>
> [3] Chen D, Chen K, Li Z, et al. Powernet: Multi-agent deep reinforcement learning for scalable powergrid control. IEEE Transactions on Power Systems, 2021, 37(2): 1007-1017.
>
> [4] Deng D, Wang J, Zhang L. Critical periodic traveling waves for a Kermack-McKendrick epidemic model with diffusion and seasonality. Journal of Differential Equations, 2022, 322: 365-395.
>
> [5] Li B, Gao S, Liang Y, et al. Estimation of regional economic development indicator from transportation network analytics. Scientific reports, 2020, 10(1): 2647.
>
> [6] Ma C, Li A, Du Y, et al. Efficient and scalable reinforcement learning for large-scale network control. Nature Machine Intelligence, 2024, 6(9): 1006-1020.
>
> [7] Zhang L, Li L, Wei W, et al. Scalable constrained policy optimization for safe multi-agent reinforcement learning. In NeurIPS 2024.

---

### Official Review · Reviewer_kFZa · 2025-11-01

**Soundness:** 3
**Presentation:** 2
**Contribution:** 3
**Rating:** 6
**Confidence:** 2

**Summary:**

This paper proposes ACR-PPO, a scalable cooperative MARL framework that enables agents to dynamically adapt their communication range under a budget. The method models decision-making as a sequential process, where a communication policy first selects a range and a behavior policy then acts based on the gathered information. Supported by a theoretical guarantee of monotonic performance improvement, experiments show that ACR-PPO offers competitive performance in networked environments.

**Strengths:**

1. The paper tackles a relevant and practical problem. Improving the scalability of communication in MARL is crucial for real-world applications, and the proposed approach of dynamically adjusting the communication range under a budget is a promising and well-motivated direction.
2. The authors provide a formal guarantee for monotonic policy improvement under communication constraints, which adds significant rigor to the proposed method.

**Weaknesses:**

1. The results are a bit difficult to interpret. For instance, while Figure 2(c) provides a valuable snapshot of the communication range changing within one episode, it would be beneficial to include more comprehensive statistics. A richer analysis might include:
    - Visualizations or statistics on how the communication range varies across different agents in a single episode. Do some agents consistently require a larger range than others?
    - An analysis of how the average communication range evolves over the entire training process (i.e., across episodes), not just within a single late-stage episode.
    - Case studies in different scenarios showing how the learned communication strategy adapts to different environmental demands.
2. The current baselines compare against full communication (CPPO) and no communication (DPPO). To make a more compelling case for the dynamic nature of ACR-PPO, it would be valuable to include a baseline with a fixed, moderate communication range. This would directly test whether dynamically adjusting the range offers a significant advantage over a well-chosen, static communication heuristic.

**Questions:**

1. How does the effectiveness of ACR-PPO vary with the underlying graph topology? For example, does the method's performance change significantly in graphs with larger diameters, larger size or different topology?
2. The paper frames the decision from the receiver's perspective, where an agent's communication policy determines the range from which it receives information. In a real-world implementation, this would seem to require a "request: phase where the receiver informs potential senders. The overall policy then becomes multi-stage: communication neighbors selection - request - communication - action policy. Does this introduce significant latency or overhead that might affect the applicability of the method? Is there any potential ways to reduce the latency?
3. Could the authors clarify the assumptions regarding agent homogeneity? Are the policies, communication budgets, and local network structures assumed to be uniform across all agents?

---

> ### Author Response · Authors · 2025-11-25
> **Reply1**
>
> We sincerely thank the reviewers for their valuable comments. We hope the responses below provide further clarification:
>
> ## W1② & W1③: An analysis...episode. / Case studies ... demands.
>
> **Experiment setting:** We report results at three times in the PowerGrid environment: Episode No.100, Episode No.400, and Episode No.800 (averaged over 3 seeds). Due to the character limit on OpenReview, the complete experimental results have been updated in the **Appendix D6 and Fig.6.**
>
> | Mean Communication Range  | Step1    | Step5   | Step11  | Step 15 | Step 19 |
> | ------------------------- | -------- | ------- | ------- | ------- | ------- |
> | ACR-PPO (No. 100 Episode) | 9.9      | 8.4     | 9.7     | 10.92   | 8.87    |
> | ACR-PPO (No. 400 Episode) | 10.2     | 9.4     | 9.7     | 8.8     | 9.8     |
> | ACR-PPO (No. 800 Episode) | **9.25** | **9.6** | **8.9** | **9.7** | **8.7** |
> | DPPO-Medium(Baseline)     | 10       | 10      | 10      | 10      | 10      |
>
> **Experiment Result:** In the early training stage, the model is limited by initialization, resulting in a large communication range. In later stages, under the influence of communication constraints, the range gradually decreases.
>
> ## W1 ① & W1③: Visualizations ... others?
>
> **Experiment setting:** The PowerGrid environment (20 agents). Similar to W②, we present the behavior of each agent at three time points: ACR-PPO (No. 100 Episode), ACR-PPO (No. 400 Episode), and ACR-PPO (No. 800 Episode), averaged over three seeds. Due to OpenReview's character limit, this figure shows only the communication patterns of odd-numbered agents under ACR-PPO (No. 800 Episode). Full results are updated in **Appendix D6 and Table 3.**
>
> | No.800 Episode | Step1               | Step5               | Step11              | Step 15             | Step 19         |
> | -------------- | ------------------- | ------------------- | ------------------- | ------------------- | --------------- |
> | Agent 1        | 17.5($\pm$ 1.5)     | 6.5($\pm$ 0.5)      | 4.5($\pm$ 3.5)      | 14($\pm$ 5)         | 9.5($\pm$ 1.5)  |
> | Agent 3        | 8.5($\pm$ 8.5)      | 4($\pm$ 4)          | 14.5($\pm$ 1.5)     | 4($\pm$ 2)          | 3.5($\pm$ 2.5)  |
> | Agent 5        | 8($\pm$ 8)          | 8($\pm$ 6)          | 14($\pm$ 1)         | 1.5($\pm$ 0.5)      | 6($\pm$ 5)      |
> | **Agent 7**    | **7.5**($\pm$ 0.5)  | **10.5**($\pm$ 3.5) | **7.5**($\pm$ 6.5)  | **6**($\pm$ 0)      | **6**($\pm$ 1)  |
> | **Agent 9**    | **13.5**($\pm$ 1.5) | **15**($\pm$ 2)     | **11.5**($\pm$ 3.5) | **14.5**($\pm$ 3.5) | **14**($\pm$ 1) |
> | Agent 11       | 8.5($\pm$ 3.5)      | 10($\pm$ 5)         | 7.5($\pm$ 4.5)      | 5($\pm$ 1)          | 10.5($\pm$ 7.5) |
> | Agent 13       | 1.5($\pm$ 1.5)      | 15($\pm$ 2)         | 2($\pm$ 2)          | 8($\pm$ 6)          | 10.5($\pm$ 1)   |
> | Agent 15       | 6.5($\pm$ 2.5)      | 13($\pm$ 6)         | 10($\pm$ 2)         | 14($\pm$ 3)         | 8.5($\pm$ 2.5)  |
> | Agent 17       | 12($\pm$ 1)         | 12.5($\pm$ 0.5)     | 10($\pm$ 8)         | 13($\pm$ 6)         | 4.5($\pm$ 3.5)  |
> | Agent 19       | 9.5($\pm$ 4.5)      | 10.5($\pm$ 1.5)     | 8($\pm$ 8)          | 8.5($\pm$ 3.5)      | 12($\pm$ 6)     |
>
> **Experiment Result.** Due to the temporary inability to visualization, as you suggested, we speculate that Agent 9 may be a highly critical node, as it consistently maintains extensive communication over prolonged periods. In contrast, Agent 7 may be a less critical or relatively independent node, as it does not require sustained in the large-scale communication.

---

> > ### Author Response · Authors · 2025-11-25
> > **Reply2**
> >
> > ## W2 & Q1：The current ... heuristic. / How does the effectiveness ... topology?
> >
> > Thank you for your suggestion. We have added a medium-communication baseline (DPPO-Medium) for each of the seven environments, including the newly added New York scenario (426 agents) (averaged over 3 seeds). Due to the character limit on OpenReview, only the results for the New York environment are shown here. The remaining experimental results have been updated **in the Appendix D5 and Fig.5.**
> >
> > **Reward in "New York scenario"**(The higher, the better)：
> >
> > |                     | 0.01M                  | 0.07M                 | 0.15M                 | 0.23M                 | 0.3M                  |
> > | ------------------- | ---------------------- | --------------------- | --------------------- | --------------------- | --------------------- |
> > | CPPO                | -10.94($\pm$ 0.47)     | -8.13($\pm$ 0.19)     | -5.74($\pm$ 0.02)     | -5.97($\pm$ 0.16)     | -3.59($\pm$ 0.45)     |
> > | DPPO                | -9.81($\pm$ 0.13)      | -10.23($\pm$ 0.50)    | -8.64($\pm$ 1.80)     | -7.09($\pm$ 1.50)     | -5.63($\pm$ 0.99)     |
> > | DPPO-Medium         | -10.80($\pm$ 0.06)     | -10.27($\pm$ 0.58)    | -7.88($\pm$ 0.02)     | -6.16($\pm$ 0.12)     | -5.26($\pm$ 0.06)     |
> > | CPPO-Regularization | -9.76($\pm$ 0.14)      | -8.11($\pm$ 0.42)     | -5.58($\pm$ 0.05)     | 5.74($\pm$ 0.17)      | -5.45($\pm$ 0.37)     |
> > | CPPO-Randomsample   | -10.64($\pm$ 0.17)     | -8.37($\pm$ 0.36)     | -6.80($\pm$ 0.21)     | -7.57($\pm$ 0.83)     | -7.12($\pm$ 0.48)     |
> > | **ACR-PPO (Ours)**  | **-11.66($\pm$ 0.39)** | **-6.81($\pm$ 0.36)** | **-6.62($\pm$ 0.33)** | **-6.37($\pm$ 0.21)** | **-5.67($\pm$ 1.35)** |
> >
> > **Cost in "New York scenario"** (The lower, the better)：
> >
> > | **Cost in "New York scenario"** | 0.01M                | 0.07M                | 0.15M                | 0.23M                | 0.3M                 |
> > | ------------------------------- | -------------------- | -------------------- | -------------------- | -------------------- | -------------------- |
> > | CPPO                            | 77.2                 | 77.2                 | 77.2                 | 77.2                 | 77.2                 |
> > | DPPO                            | 0                    | 0                    | 0                    | 0                    | 0                    |
> > | DPPO-Medium                     | 7.84                 | 7.84                 | 7.84                 | 7.84                 | 7.84                 |
> > | CPPO-Regularization             | 71.52($\pm$ 2.65)    | 72.22($\pm$ 3.96)    | 75.85($\pm$ 0.88)    | 72.83($\pm$ 4.36)    | 74.20($\pm$ 0.26)    |
> > | CPPO-Randomsample               | 7.61($\pm$ 3.63)     | 8.61($\pm$ 2.53)     | 6.49($\pm$ 1.48)     | 7.24($\pm$ 3.16)     | 8.36($\pm$ 3.36)     |
> > | **ACR-PPO (Ours)**              | **0.50($\pm$ 0.03)** | **0.47($\pm$ 0.02)** | **0.42($\pm$ 0.03)** | **0.38($\pm$ 0.04)** | **0.41($\pm$ 0.02)** |
> >
> > **Experiment Result:**  In terms of performance and cost, our method achieves adaptive communication range selection and still maintains certain advantages over a policy with a medium communication range (DPPO-Medium) in large-scale scenario.
> > ## Q2：The paper ... latency?
> >
> > ACR-PPO's communication policy network is small and simple. Thus, despite the overhead from training this network, ACR-PPO's total training time is not significantly increased compared to DPPO (which lacks convergence guarantees). Although CPPO has the shortest runtime by not simulating actual communication, real-world centralized training faces significant pressure from extensive communication demands leading to time delay and packet loss. Our method addresses these substantial overheads by learning communication policies, which is a valuable trade-off.
> >
> > ## Q3: Could the ... agents?
> >
> > We sincerely apologize for the confusion caused by this issue. Due to constraints on computational resources and training time, we adopted parameter sharing, which is a common practice in the MARL community [1][2] to implement our method. However, we would like to emphasize that, in real-world deployments or under sufficient computational resources, our method can naturally be extended to heterogeneous agents without any fundamental limitations.
> >
> > ## Reference
> >
> > [1] Ma C, Li A, Du Y, et al. Efficient and scalable reinforcement learning for large-scale network control. Nature Machine Intelligence, 2024, 6(9): 1006-1020.
> >
> > [2] Zhang L, Li L, Wei W, et al. Scalable constrained policy optimization for safe multi-agent reinforcement learning. In NeurIPS 2024.

---

### Official Review · Reviewer_VKmD · 2025-11-01

**Soundness:** 3
**Presentation:** 2
**Contribution:** 3
**Rating:** 6
**Confidence:** 2

**Summary:**

This paper proposes ACR-PPO, an adaptive communication range policy optimization algorithm for cooperative multi-agent reinforcement learning (MARL) under communication cost constraints. The approach formulates the optimization process as a sequential two-policy update mechanism: communication policy and behavior policy. The idea is intuitive and make sense. The authors have provided detailed theoretical proofs and derivations, along with experimental results across multiple benchmarks, which validate the effectiveness of the proposed method.

**Strengths:**

1.The work directly tackles an important bottleneck in scalable MARL—the exponential growth of inter-agent communication cost—and proposes a mechanism that can flexibly adapt to dynamic and resource-constrained environments.

2.Theoretical Guarantees: The paper provides well-developed analysis, which enhances the credibility of the proposed method.

3.Strong Empirical Validation: Quantitative results demonstrate that ACR-PPO can adaptively trade off performance and cost, often achieving comparable performance with dramatically reduced communication.

**Weaknesses:**

1. While this work achieves comparable performance with dramatically reduced communication via a sequential process, it is common sense that such sequential strategies typically involve a trade-off between time and space. If feasible, the authors should provide additional comparative experiments related to time (frequency) to clarify the overhead incurred. In my opinion, controlling frequency (time delay) should also be a critical factor for the real-world performance of multi-agent systems.

2. The goal of this paper is scalability (see Abstract, Introduction, Section 4). While the authors’ approach shows promising potential, the compared methods do not seem to be tested in larger-scale environments. If there exist experimental setups for larger scenarios in this field, the authors should supplement such experiments. Large-scale experiments are not as straightforward as theoretical verification; they typically involve more factors that need to be considered, such as communication latency.  Without experiments in such scenarios, the authors’ claims about scalability are weakened-—a topic frequently discussed in the other domains.

3. Given my limited familiarity with this field, I am not certain about the reasonableness of the selected baselines. However, I notice that only one 2025 paper is cited for comparison. If there exist more recent relevant works, the authors should typically include performance comparisons with them.

4. The paper is somewhat overly focused on theoretical rigor, and the authors are encouraged to provide more implementation details. Regarding the theoretical section, the authors are also advised to offer more intuitive explanations—this would facilitate quick comprehension for researchers from other fields who may be interested in the work.

**Questions:**

Please see Weaknesses.

---

> ### Author Response · Authors · 2025-11-25
> **Reply1**
>
> We sincerely appreciate the reviewers’ thoughtful comments and hope the responses below provide further clarification.
> ## W1: While this work ... systems.
>
> In this paper, we derive a theorem on the monotonic improvement of joint policies under low-communication conditions based on the spatial correlation decay assumption and trust region theory, and propose an implementation scheme for sequentially updating the communication policy network and the behavior policy network. It is worth noting that the communication policy network is a small and simple network(**as shown in Appendix D3**). Therefore, although training the communication policy network incurs some time overhead (**as shown in Table 2**), the training time of ACR-PPO does not increase significantly compared to DPPO, which operates without communication but theoretically lacks convergence guarantees and performance advantages. Additionally, since we do not simulate the actual communication process between agents or between agents and a central server, CPPO achieves the shortest runtime. However, in real-world scenarios, the demand for extensive communication, along with issues such as time delay and packet loss caused by dense communication, imposes significant pressure on centralized training. Our work aims to mitigate these substantial overheads by learning communication policies, which is often a worthwhile endeavor.
> ## W2: The goal of this paper ... domains.
>
> ① **Regarding larger-scale experiments:** We have added experiments on the New York environment, which simulates traffic light control across New York City with 436 agents. This scenario is widely used in existing scalable MARL methods [1] [2]. Due to OpenReview constraints, we include only a summary table here. The full experimental setup and learning curves for the New York environment have been updated in the **Appendix D5**.
>
> **Reward in "New York scenario"**(The higher, the better)：
>
> |                     | 0.01M                  | 0.07M                 | 0.15M                 | 0.23M                 | 0.3M                  |
> | ------------------- | ---------------------- | --------------------- | --------------------- | --------------------- | --------------------- |
> | CPPO                | -10.94($\pm$ 0.47)     | -8.13($\pm$ 0.19)     | -5.74($\pm$ 0.02)     | -5.97($\pm$ 0.16)     | -3.59($\pm$ 0.45)     |
> | DPPO                | -9.81($\pm$ 0.13)      | -10.23($\pm$ 0.50)    | -8.64($\pm$ 1.80)     | -7.09($\pm$ 1.50)     | -5.63($\pm$ 0.99)     |
> | DPPO_Medium         | -10.80($\pm$ 0.06)     | -10.27($\pm$ 0.58)    | -7.88($\pm$ 0.02)     | -6.16($\pm$ 0.12)     | -5.26($\pm$ 0.06)     |
> | CPPO-Regularization | -9.76($\pm$ 0.14)      | -8.11($\pm$ 0.42)     | -5.58($\pm$ 0.05)     | 5.74($\pm$ 0.17)      | -5.45($\pm$ 0.37)     |
> | CPPO-Randomsample   | -10.64($\pm$ 0.17)     | -8.37($\pm$ 0.36)     | -6.80($\pm$ 0.21)     | -7.57($\pm$ 0.83)     | -7.12($\pm$ 0.48)     |
> | **ACR-PPO (Ours)**  | **-11.66($\pm$ 0.39)** | **-6.81($\pm$ 0.36)** | **-6.62($\pm$ 0.33)** | **-6.37($\pm$ 0.21)** | **-5.67($\pm$ 1.35)** |
>
> **Cost in "New York scenario"** (The lower, the better)：
>
> | **Cost in "New York scenario"** | 0.01M                | 0.07M                | 0.15M                | 0.23M                | 0.3M                 |     |
> | ------------------------------- | -------------------- | -------------------- | -------------------- | -------------------- | -------------------- | --- |
> | CPPO                            | 77.2                 | 77.2                 | 77.2                 | 77.2                 | 77.2                 |     |
> | DPPO                            | 0                    | 0                    | 0                    | 0                    | 0                    |     |
> | DPPO-Medium                     | 7.84                 | 7.84                 | 7.84                 | 7.84                 | 7.84                 |     |
> | CPPO-Regularization             | 71.52($\pm$ 2.65)    | 72.22($\pm$ 3.96)    | 75.85($\pm$ 0.88)    | 72.83($\pm$ 4.36)    | 74.20($\pm$ 0.26)    |     |
> | CPPO-Randomsample               | 7.61($\pm$ 3.63)     | 8.61($\pm$ 2.53)     | 6.49($\pm$ 1.48)     | 7.24($\pm$ 3.16)     | 8.36($\pm$ 3.36)     |     |
> | **ACR-PPO (Ours)**              | **0.50($\pm$ 0.03)** | **0.47($\pm$ 0.02)** | **0.42($\pm$ 0.03)** | **0.38($\pm$ 0.04)** | **0.41($\pm$ 0.02)** |     |
>
> **Experiment Result:** In terms of performance and cost, our method achieves adaptive communication range selection and still maintains certain advantages in large-scale scenario.

---

> ### Author Response · Authors · 2025-11-25
> **Reply2**
>
> ## W3：Given my limited ... them.
> ① **Regarding the Discussion on Baselines.**
> Current MARL training methods fail in networked, large-scale systems due to two fundamental limitations:
> - Centralized Methods (e.g., CTDE/CTCE): These rely on the **global state-action value function** $Q(\mathbf{s}, \mathbf{a})$. Methods like Gated-ACML [3] and AC2C [4] still rely on this global information, incurring high communication costs that are impractical for our large-scale networked setting.
> - Fully Decentralized Methods (DTDE): These rely solely on the **local state-action value function** $Q_i(s_i, a_i)$, suffering from non-stationarity and degraded performance due to a lack of necessary coordination.
>
> Our work addresses the Networked MARL setting where agents communicate only with neighbors within a $\kappa$-hop range [1][5][6]. Under the spatial exponential decay assumption, we leverage a **truncated value function** $Q(\mathbf{s}^{\kappa}, \mathbf{a}^{\kappa})$, which uses only local and $\kappa$-hop neighbor information. This allows for scalability while retaining necessary cooperation. This approach sits uniquely between the centralized and fully decentralized extremes. Although this enables local cooperation, determining an appropriate communication range remains an open problem. Existing methods either fix the communication range [1][5][6] or sample from a fixed distribution [7], both of which exhibit limited adaptability when the environment changes.
>
> ② **Additional Experiments:** We further add a regularization-based method, which adding an L1 penalty on the sum of each agent’s feature representations. The regularization results on other maps have also been added and updated (**such as: Fig 1**). However, due to the nature of L1 regularization, which only encourages sparsity by reducing the magnitude of communication content without explicitly enforcing zero communication, it fails to provide a precise or tight constraint on actual communication costs. As a result, this approach is insufficient for directly controlling or bounding communication expenditure.
>
> ## W4: The paper is somewhat ... work.
>
> We apologize for any confusion caused by the complexity of our approach. We hereby provide the following clarification:
>
> ① **Regarding implementation details**, we have provided a more comprehensive description in **Appendix D.3**.
>
> ② **Regarding the theoretical section.**
> To address the problem of communication range selection in Networked MARL, we formulate this work as a constrained reinforcement learning problem. **First,** based on the assumptions of spatial decay (Assumption 1 and Assumption 2), we define the truncation function and extend it to the time-varying case (Definition 2). **Then,** we derive the corresponding forms of the time-varying $Q$, $V$, $A$ function (Proposition 1, Corollary 1, Corollary 2). Based on these, we establish a performance lower bound for monotonic policy improvement (Proposition 2). Similarly, for the cost constraint, we define the surrogate objective for communication cost (Lemma 2). **Finally,** by jointly maximizing the lower bound in Proposition 2 and minimizing the upper bound in Lemma 2, we arrive at Theorem 1, which guarantees that agent performance can be monotonically improved while keeping the communication range within the specified constraints.
> ## Reference
>
> [1] Ma C, Li A, Du Y, et al. Efficient and scalable reinforcement learning for large-scale network control. Nature Machine Intelligence, 2024, 6(9): 1006-1020.
>
> [2] Duan W, Lu J, Xuan J. Bayesian Ego-graph inference for Networked Multi-Agent Reinforcement Learning. In NeurIPS 2025.
>
> [3] Mao H, Zhang Z, Xiao Z, et al. Learning agent communication under limited bandwidth by message pruning. In AAAI 2020.
>
> [4] Wang X, Li X, Shao J, et al. AC2C: Adaptively controlled Two-Hop Communication for Multi-Agent Reinforcement Learning. In AAMAS 2023.
>
> [5] Zhang L, Li L, Wei W, et al. Scalable constrained policy optimization for safe multi-agent reinforcement learning. In NeurIPS 2024.
>
> [6] Liang H, Shi S, Zhang Y, et al. Causality Meets Locality: Provably Generalizable and Scalable Policy Learning for Networked Systems. In arXiv 2025.
>
> [7] Leopoldo A, Sean V, Santiago P, et al. Cooperative multi-agent assignment over stochastic graphs via constrained reinforcement learning. In arXiv 2025.

---

### Comment · Area_Chair_Qdsf · 2025-11-28

Dear reviewers,

Please check authors' responses and provide your feedback.

AC

---

### Author Response · Authors · 2025-11-29
**Global Response**

Dear PCs, SACs, ACs, Reviewers:

We sincerely thank all reviewers for the time and effort dedicated to reviewing our paper, as well as for their insightful and constructive feedback. These comments have significantly enhanced the clarity, rigor, and breadth of our work.

## First, we particularly appreciate the reviewers’ recognition of several strengths of our method, including:

- “Appreciation for the well-motivated and clearly structured approach” (Reviewers: VKmD, kFZa, 4vHw);
- “Acknowledgment of the theoretical guarantees” (Reviewers: VKmD, kFZa, jnKk, 4vHw);
- “Diverse and interesting applications” (Reviewers: VKmD, jnKk);
- “Relevance and  interest to researchers in the MARL with communication community” (Reviewer: 4vHw).

## Second, we have carefully addressed all reviewer comments in turn and implemented the necessary revisions in the following sections of the new submission:
In response to the reviewers’ comments and concerns, we have provided detailed, point-by-point replies to each reviewer and substantially revised the manuscript, adding or updating **5 figures/tables, 4 main-text sections, and 8 Appendix sections**. The key revisions are summarized as follows:

1. **Motivation**:  In response to Reviewer 4vHw, we clarified the Networked MARL setting and differentiated our approach from prior work from the perspective of the global state (**Appendix B.1**).
2. **Significance**: Significance: In real-world multi-agent scenarios, each agent typically operates in a partially observable and non-stationary environment. This paper aims to ensure theoretical guarantees for monotonic improvement in joint policy performance, while minimizing the need for extensive and dense communication, which often leads to issues such as time delay, packet loss, and scalability bottlenecks in larger multi-agent systems. (**Section 4.2**)
3. **Theoretical Derivation**:  Addressing concerns raised by Reviewers jnKk and 4vHw regarding insufficient theoretical grounding, we added more detailed assumptions and descriptions based on spatial decay (**Section 2; Appendices B.1 and B.2**).
4. **Experiments**:  Following suggestions from Reviewers VKmD and kFZa, we added five new sets of experiments:
	1. A large-scale scenario "New York" with 436 agents to verify scalability (**Appendix D.5**). Results show our method maintains performance while reducing communication cost even at this scale.
	2. Analysis of per-agent communication ranges within a single episode (**Appendix D.6**), revealing the existence of “key agents” with consistently long-range communication and “peripheral agents” with consistently short-range communication.
	3. Evaluation of mean communication ranges across episodes (**Appendix D.6**), demonstrating that our method progressively reduces overall communication cost over training.
	4. An additional baseline, DPPO-Medium, where all agents communicate with a fixed medium-range neighborhood (**across all environments**). Results show our method achieves comparable performance with lower communication cost.
	5. Runtime analysis across all environments, showing that the computational overhead of our communication policy is only marginally higher than baselines (**Appendix D.3**).
5. **Miscellaneous clarifications and textual refinements:** throughout the paper.

We hope that our detailed responses and the substantial revisions to the paper satisfactorily address your concerns.

---

### Meta-Review · Area_Chair_8Qo4 · 2025-12-20

**Summary:**

The paper tackles scalable cooperative MARL by learning an adaptive communication range under a cost budget, and the revised version strengthens the empirical evidence. However, the decision is dominated by an unresolved theoretical correctness concern raised with high confidence: the proof of Lemma 1 hinges on assuming that the two induced visitation distributions match for a number of steps proportional to the truncation range. The revision mainly adds explanation rather than providing a corrected statement/proof. Since Lemma 1 underpins the paper’s key theoretical claims about truncation error control and associated guarantees, I view this as a substantial risk that prevents me from recommending acceptance at this stage, despite the otherwise promising empirical results and relevance of the problem.

**Reviewer Concerns:**

1. Lemma 1 correctness/consistency
2. Related work / baseline coverage

**Reviewer Scores:**

VKmD: 6
kFZa: 6
4vHw：6
jnKk：2

---

### Decision · Program_Chairs · 2026-01-26

Reject